# Simple Imputation Rules for Prediction with Missing Data: Theoretical Guarantees vs. Empirical Performance

**Dimitris Bertsimas**                                                    *dbertsim@mit.edu*
*Sloan School of Management*
*Massachusetts Institute of Technology, Cambridge (MA), USA*

**Arthur Delarue**                                        *arthur.delarue@isye.gatech.edu*
*H. Milton Stewart School of Industrial and Systems Engineering*
*Georgia Institute of Technology, Atlanta (GA), USA*

**Jean Pauphilet**                                                  *jpauphilet@london.edu*
*London Business School, London, UK*

**Reviewed on OpenReview:** *https://openreview.net/forum?id=IKH5ziX9dk*

## Abstract

Missing data is a common issue in real-world datasets. This paper studies the performance of impute-then-regress pipelines by contrasting theoretical and empirical evidence. We establish the asymptotic consistency of such pipelines for a broad family of imputation methods. While common sense suggests that a 'good' imputation method produces datasets that are plausible, we show, on the contrary, that, as far as prediction is concerned, crude can be good. Among others, we find that mode-impute is asymptotically sub-optimal, while mean-impute is asymptotically optimal. We then exhaustively assess the validity of these theoretical conclusions on a large corpus of synthetic, semi-real, and real datasets. While the empirical evidence we collect mostly supports our theoretical findings, it also highlights gaps between theory and practice and opportunities for future research, regarding the relevance of the MAR assumption, the complex interdependency between the imputation and regression tasks, and the need for realistic synthetic data generation models.

## 1 Introduction

Real-world datasets are plagued with missing values. For inference purposes, the key assumption is that data entries are missing *at random* (MAR)—i.e., the fact that a feature is missing and its (unobserved) value are independent, conditional on the observed features (Rubin, 1976). Under the MAR assumption, valid inference can be performed by using tailored EM algorithms that can handle missing values (e.g., Dempster et al., 1977; Ibrahim et al., 2005; Jiang et al., 2020). Alternatively, one can impute the missing values, estimate the parameter of interest on the imputed dataset, and obtain valid confidence intervals by accounting for imputation errors (e.g., via multiple imputations as in Rubin, 1987). A myriad of imputation techniques have been proposed based on mean and mode imputation (Little & Rubin, 2019), $k$-nearest neighbors (Troyanskaya et al., 2001; Brás & Menezes, 2007), least square regression (Bø et al., 2004; Kim et al., 2005; Cai et al., 2006; Zhang et al., 2008), support vector machine/regression (Wang et al., 2006; Bertsimas et al., 2018), decision trees (Bertsimas et al., 2018), neural networks (Yoon et al., 2018), or factor analysis and other dimension reduction techniques (Mohamed et al., 2009; Husson et al., 2019). All these methods assume that the MAR assumption holds, hence missing values can be guessed accurately from non-missing values. However, without any further assumptions about the data, the MAR assumption cannot be tested nor refuted from the data (see Little, 1988; Jaeger, 2006, for some additional assumptions amenable to statistical testing). In addition, most inference guarantees become invalid as soon as the data is *not* missing at random (NMAR).

Prediction, however, is a different task from inference (Shmueli, 2010) that requires a different treatment for missing data. Besides tree-based methods that can natively handle missing values (see Josse et al., 2019, Section 6 and references therein), prediction with missing data is usually performed following a similar approach as the one for inference, namely an *impute-then-regress* approach in which one first imputes missing values, and then trains a model on the imputed dataset. Accordingly, the MAR assumption, which is required for the imputation step to be unbiased, is often perceived as a requirement for impute-then-regress approaches.

Our paper contributes to the recent literature on the analysis of impute-then-regress pipelines and the relevance of the MAR assumption. Our analysis starts by a theoretical guarantee on the asymptotic consistency of impute-then-regress pipelines for a broad class of imputation methods, which generalizes (Josse et al., 2019) and complements (Le Morvan et al., 2020b; 2021) the existing literature. To the best of our knowledge, our result is the first to cover the imputation of categorical features, and not only continuous ones. In particular, our result characterizes the (asymptotically) "optimal" imputation methods as those that encode for missingness as obviously as possible. For simple imputation rules, conclusions from the theory are thus clear: mode imputation (though often used in practice, see Jäger et al., 2021) is sub-optimal, while mean-imputation is asymptotically optimal. We then exhaustively assess the validity of these conclusions on synthetic, semi-real, and real datasets, with the double objective of supporting our theoretical findings with empirical evidence as well as eliciting gaps between theory and practice that could motivate future research in the area.

The rest of the paper is organized as follows:

- We study the consistency of generic imputation rules in the infinite-data regime in Section 2. While inference requires the imputed values to be unbiased estimates of the missing entries (a property usually satisfied under MAR), we show that, for prediction, impute-then-regress works if the imputed value codifies missingness, i.e., if the imputed values are as conspicuous as possible. Our analysis applies to continuous as well as categorical missing features and to a broad family of imputation rules, including mode and mean imputation.

- For mode and mean imputation, we contrast these theoretical consistency results with their empirical performance on a large corpus of synthetic, semi-real (i.e., real-world design matrix and missingness patterns but synthetic signals), and real-world datasets.

  - For discrete variables (Section 3.2), our theoretical analysis suggests that mode imputation cannot lead to consistent predictions and that encoding missingness as its own category should be preferred. Empirical evidence on synthetic and semi-real datasets strongly supports the validity of these findings, although the evidence on real data is not conclusive.

  - For continuous variables (Section 3.3), rules as simple as mean imputation are theoretically consistent. We compare the performance of mean imputation with non-linear iterative imputation methods (van Buuren & Groothuis-Oudshoorn, 2010). As suggested by theory, we observe no clear downside from using mean imputation on average. However, we observe that the missingness mechanism, the fraction of missing entries, and the complexity of the downstream predictive model have a significant impact on which imputation model performs best.

- Finally, we discuss the limitations of our theoretical and empirical findings in Section 4 with the hope to guide future research. In particular, the contrast between theory and practice highlights interesting directions related with the relevance of the MAR assumption in predictive setting; the complex interactions between the imputation and the predictive models and its impact on finite-sample performance; and the need for realistic generative models.

**Notation.** We denote scalars by lowercase characters ($x$) and random variables by uppercase characters ($X$). Boldfaced characters denote vectors (e.g., $\boldsymbol{x}$ is a vector and $\boldsymbol{X}$ is a random vector). The symbol $\perp\!\!\!\perp$ designates independent random variables. For any positive integer $n$, let $[n] = \{1, \dots, n\}$.

**Code.** All code used to implement and evaluate imputation strategies is available on Github.[1]

## 2 Theoretical Consistency with Generic Imputation Rules

### 2.1 Setting

**Generalities and notation.** We consider the task of predicting a target or dependent variable, modeled as a random variable $Y$, from a set of features, modeled as a random vector $\boldsymbol{X}$. For each feature $X_i$, the binary random variable $M_i$ indicates whether it is missing (1) or observed (0). Our predictive model is trained using data: $n$ i.i.d. samples $(\boldsymbol{x}_i, \boldsymbol{m}_i, y_i)$, $i \in [n]$, where $\boldsymbol{x}_i \in \mathbb{R}^d$ is the vector of covariates and $\boldsymbol{m}_i \in \{0,1\}^d$ is a vector indicating the missing covariates—$m_{ij} = 1$ if $x_{ij}$ is missing, 0, otherwise— and $y_i \in \mathbb{R}$ is the output of interest. For every data point $i$, $\|\boldsymbol{m}_i\|_0 := \sum_j m_{ij}$ covariates are missing. We refer to $\boldsymbol{m}_i$ as the missingness *indicator* or missingness *pattern* of sample $i$. We further denote $\boldsymbol{o}(\boldsymbol{x}_i, \boldsymbol{m}_i)$ the $(d - \|\boldsymbol{m}_i\|_0)$-dimensional vector of **o**bserved covariates (Seaman et al., 2013).

For the task of predicting $Y$ given covariates $\boldsymbol{X}$, a predictor $\hat{f}_n$ trained on a dataset of $n$ observations is called *(asymptotically) consistent* if $\lim_{n\to\infty} \hat{f}_n(\boldsymbol{x}) = \mathbb{E}[Y|\boldsymbol{X} = \boldsymbol{x}]$. Moreover, $\hat{f}_n$ is *universally* consistent if the previous statement holds for any distribution of $(\boldsymbol{X}, Y)$.

**Objective.** Our goal in this section is to investigate the asymptotic consistency of impute-then-regress strategies for a broad family of imputation rules.

For ease of exposition, we consider a simplified setting in $d$ dimensions where only the first covariate $X_1$ is missing. The optimal (consistent) predictor for the target variable $Y$ is

$$\begin{cases} \mathbb{E}[Y|\boldsymbol{X} = \boldsymbol{x}, M_1 = 0], & \text{if } m_1 = 0, \\ \mathbb{E}[Y|\boldsymbol{X}_{2:d} = \boldsymbol{x}_{2:d}, M_1 = 1], & \text{if } m_1 = 1. \end{cases} \tag{1}$$

However, like in Josse et al. (2019), our analysis could be extended to the case with more than one missing covariate. Here, we concisely denote $\boldsymbol{x}_{2:d}$ the $(d-1)$-dimensional vector $(x_2, \ldots, x_d)$.

We study the common practice of imputing a deterministic value for $X_1$ whenever it is missing. In particular, we allow for the imputed value to be a deterministic function of the other covariates, denoted $\mu(\boldsymbol{x}_{2:d})$. This model captures mean and mode imputation, as well as conditional mean and mode imputation. For clarity, we will denote $\boldsymbol{X}^\mu$ the random variable obtained from $(\boldsymbol{X}, \boldsymbol{M})$ after $\mu$-imputation, i.e., $X_1^\mu = X_1$ if $M_1 = 0$, $X_1^\mu = \mu(\boldsymbol{X}_{2:d})$ if $M_1 = 1$, and $\boldsymbol{X}_{2:d}^\mu = \boldsymbol{X}_{2:d}$.

### 2.2 Asymptotic prediction function

We now introduce the main result of this section, which characterizes the asymptotic performance of impute-then-regress strategies based on deterministic imputation rules.

**Theorem 2.1.** *Consider a universally consistent learning algorithm when trained on any fully observed dataset. Imputing $\mu(\boldsymbol{x}_{2:d})$ for $X_1|\boldsymbol{X}_{2:d} = \boldsymbol{x}_{2:d}$ when $X_1$ is missing ($M_1 = 1$) on the training set and training a predictor on the imputed dataset leads, in the limit with infinite data, to the following prediction rule, denoted $f_{\mu\text{-impute}}(\boldsymbol{x})$ and equal almost everywhere to*

$$f_{\mu\text{-impute}}(\boldsymbol{x}) = \begin{cases} \mathbb{E}[Y|\boldsymbol{X} = \boldsymbol{x}, M_1 = 0], & \text{if } x_1 \neq \mu(\boldsymbol{x}_{2:d}), \\ \alpha(\boldsymbol{x})\,\mathbb{E}[Y|\boldsymbol{X}_{2:d} = \boldsymbol{x}_{2:d}, M_1 = 1] \\ \qquad + (1 - \alpha(\boldsymbol{x}))\mathbb{E}[Y|X_1 = \mu(\boldsymbol{x}_{2:d}), \boldsymbol{X}_{2:d} = \boldsymbol{x}_{2:d}, M_1 = 0], & \text{otherwise.} \end{cases}$$

*where*

- $\eta(\boldsymbol{x}) = \mathbb{P}(M_1 = 1|\boldsymbol{X}_{2:d} = \boldsymbol{x}_{2:d})$ *is the probability that $X_1$ is missing given $\boldsymbol{X}_{2:d}$,*

---

[1]https://github.com/adelarue/PMD

- $p_\mu(\boldsymbol{x}) = \mathbb{P}(X_1 = \mu(\boldsymbol{x}_{2:d}), M_1 = 0 | \boldsymbol{X}_{2:d} = \boldsymbol{x}_{2:d})$ *is the probability for the true $X_1$ to take the imputed value $\mu(\boldsymbol{x}_{2:d})$ and not be missing, given the other covariates,*

- *and $\alpha(\boldsymbol{x}) = \dfrac{\eta(\boldsymbol{x})}{\eta(\boldsymbol{x}) + p_\mu(\boldsymbol{x})}$ is the posterior probability that $X_1$ was missing before imputation, given that it takes the value $\mu(\boldsymbol{x}_{2:d})$ after imputation, i.e., $\mathbb{P}(M_1 = 1 | X_1^\mu = \mu(\boldsymbol{x}_{2:d}), \boldsymbol{X}_{2:d} = \boldsymbol{x}_{2:d})$.*

Theorem 2.1 characterizes the effect of an arbitrary deterministic imputation rule on the downstream prediction rule in the infinite-data setting. Consider applying $f_{\mu\text{-impute}}$ to a new observation $\boldsymbol{x}$. Either $x_1 \neq \mu(\boldsymbol{x}_{2:d})$ or $x_1 = \mu(\boldsymbol{x}_{2:d})$. Recall that in the infinite-data regime, only the training data in the neighborhood of $\boldsymbol{x}$ affects the prediction. In the first case, $\boldsymbol{x}$ is almost surely surrounded by points with no missing entries, so $f_{\mu\text{-impute}}$ predicts the conditional expectation $\mathbb{E}[Y | \boldsymbol{X} = \boldsymbol{x}, M_1 = 0]$. In the second case, however, the points in the neighborhood of $\boldsymbol{x}$ come from a *mixture* of two distributions, depending on whether $x_1 = \mu(\boldsymbol{x}_{2:d})$ occurs artificially due to imputation, or naturally in the data: some points have a missing first feature imputed by $\mu(\cdot)$, and therefore are sampled from $(\mu(\boldsymbol{x}_{2:d}), \boldsymbol{X}_{2:d}) | M_1 = 1$; other points are sampled from the original distribution $(X_1, \boldsymbol{X}_{2:d}) | M_1 = 0$. The predicted outcome is a weighted average of both conditional expectations, with $\alpha(\boldsymbol{x})$ being the proper weighting factor. We defer the detailed proof to Appendix A.

Interestingly, we observe that $\alpha(\boldsymbol{x}) \neq \eta(\boldsymbol{x}) = \mathbb{P}(M_1 = 1 | \boldsymbol{X}_{2:d} = \boldsymbol{x}_{2:d})$ in general. Even when $X_1$ and $M_1$ are conditionally independent (i.e., MAR assumption), imputation induces correlation between $X_1^\mu$ and $M_1$.

We note that Theorem 2.1 generalizes Josse et al. (2019, Theorem 4), which only applies to constant imputation for continuous features and requires the MAR assumption. Theorem 2.1 (or its implications for asymptotic consistency in Corollary 2.2) can also be viewed as a weaker version of Le Morvan et al. (2021, Theorem 3.1) since it applies to a more restrictive setting with only a single missing covariate. However, Le Morvan et al. (2021, Theorem 3.1) applies to continuous features, while our result equally applies to discrete features. Theorem 2.1 elicits an explicit condition on the imputation function (through the quantity $\alpha(\boldsymbol{x})$) that drives asymptotic consistency, hence providing further intuition on what permits an impute-then-regress pipeline to learn the Bayes-optimal predictor.

## 2.3 Implications of Theorem 2.1

We now apply Theorem 2.1, to study the out-of-sample predictions from a learner trained on $\mu$-imputed data. For a new observation $(\boldsymbol{o}(\boldsymbol{x}, m_1), m_1)$, we apply $\mu$-imputation and then predict according to $f_{\mu\text{-impute}}(\boldsymbol{x}^\mu)$. For the sake of the discussion, we assume that $\mathbb{E}[Y | \boldsymbol{X}_{2:d} = \boldsymbol{x}_{2:d}, M_1 = 1] \neq \mathbb{E}[Y | X_1 = \mu(\boldsymbol{x}_{2:d}), \boldsymbol{X}_{2:d} = \boldsymbol{x}_{2:d}, M_1 = 0]$, otherwise the imputation is obviously harmless.

If, on one hand, $x_1$ is not originally missing ($m_1 = 0$), then $x_1^\mu = x_1$ and the impute-then-predict rule agrees with the Bayes-optimal predictor, $\mathbb{E}[Y | \boldsymbol{X} = \boldsymbol{x}, M_1 = 0]$, almost everywhere, if either $x_1 \neq \mu(\boldsymbol{x}_{2:d})$ almost surely ($p_\mu(\boldsymbol{x}) = 0 \Leftrightarrow \alpha(\boldsymbol{x}) = 1$) or $\alpha(\boldsymbol{x}) = 0$. On the other hand, if $m_1 = 1$, then $x_1^\mu = \mu(\boldsymbol{x}_{2:d})$, so the impute-then-predict rule agrees with the Bayes-optimal estimator $\mathbb{E}[Y | \boldsymbol{X}_{2:d} = \boldsymbol{x}_{2:d}, M_1 = 1]$ if and only if $\alpha(\boldsymbol{x}) = 1$.

Again, $\alpha(\boldsymbol{x})$ corresponds to the posterior probability that $X_1$ was missing in the original observation given that $X_1^\mu = \mu(\boldsymbol{x}_{2:d})$. In other words, Theorem 2.1 indicates that consistency is achieved as long as the predictor can almost surely de-impute, that is properly guess after imputation whether $X_1$ was originally missing or not. This discussion can be summarized by the following corollary:

**Corollary 2.2.** *Under the assumptions and notations of Theorem 2.1, $\mu$-imputation-then-regress asymptotically (i.e., in the infinite-data regime) leads to Bayes-optimal estimates at $\boldsymbol{x}$ if and only if $\alpha(\boldsymbol{x}) = 1$ or $\mathbb{E}[Y | \boldsymbol{X}_{2:d} = \boldsymbol{x}_{2:d}, M_1 = 1] = \mathbb{E}[Y | X_1 = \mu, \boldsymbol{X}_{2:d} = \boldsymbol{x}_{2:d}, M_1 = 0]$.*

Despite the simplicity of the underlying intuition, Theorem 2.1 challenges common practice. Indeed, one could think that a 'good' imputation method should produce datasets that are plausible, i.e., where imputed and non-imputed observations are indistinguishable ($\alpha(\boldsymbol{x}) = 0$). On the contrary, as far as predictive power is concerned, Theorem 2.1 speaks in favor of imputation methods that can be almost surely de-imputed ($\alpha(\boldsymbol{x}) = 1$), because they can be used as an encoding for missingness. In short, as far as prediction is

concerned, crude is good. This conclusion also supports the common practice of adding $\boldsymbol{m}$ as part of the predictive features, a simple but powerful idea which is gaining traction in the deep learning community (Van Ness et al., 2023; Van Ness & Udell, 2023).

# 3 Empirical Validation

## 3.1 Setting

Much of the literature on missing data relies on synthetic data for validation, mostly because it grants the experimenter full knowledge of the missing values themselves. However, as a result, missing data patterns may not match those found in the real world. In this paper, we try to bridge the gap between real and synthetic data by creating a diverse corpus of synthetic, semi-real, and real datasets. We briefly summarize our methodology here; more details can be found in Appendix C. For the synthetic datasets, we generate the design matrix $\boldsymbol{X}$ with $n$ observations (ranging from 40 to 1,000) and synthetic signals of the form $Y = f(\boldsymbol{X}) + \epsilon$, where the function $f$ is either linear or the output of a two-layer neural network (NN). We ampute (i.e., drop) data either completely at random (MCAR) or by censoring extreme values (which is a special case of NMAR), and consider 8 different proportions of missing entries. In total, we generate 49 datasets for each of the $2 \times 2 \times 8 = 32$ configurations. For the semi-real and real datasets, we assemble a corpus of 63 publicly available datasets with missing data, from the UCI Machine Learning Repository and the RDatasets Repository. We consider the design matrix and the missingness patterns from the real datasets. For semi-real instances, we generate synthetic response variables $Y$ (again, according to a linear or neural network model). We control how $Y$ depends $\boldsymbol{X}$ and $\boldsymbol{M}$, and consider relationships corresponding to missing at random (MAR), not missing at random (NMAR) or adversially missing (AM). Experiments on each dataset are replicated 10 times, with different training/test splits.

## 3.2 Categorical variables: mode imputation is inconsistent

**Theory.** For discrete features, choosing $\mu$ as the mode of the distribution of $X_1|M_1 = 0$ (a.k.a. mode imputation) is one of the advised methods in practice. However, with this choice of $\mu$, for any $\boldsymbol{x}$, we have $p_\mu(\boldsymbol{x}) > 0$ so $\alpha(\boldsymbol{x}) < 1$ and mode impute-then-regress cannot be asymptotically consistent by Corollary 2.2. Conversely, choosing $\mu$ outside of the original support of $X_1$, i.e., encoding missingness as a new category/value, provides consistency.

The practical implications of Corollary 2.2 for categorical variables are thus clear: We should encode missingness as a new category instead of imputing the mode. Accordingly, we numerically compare the out-of-sample performance of these two approaches.

**Experimental setting.** We evaluate the performance of mode imputation on our synthetic, semi-real, and real datasets. For the semi-real and real instances, we consider the 41 datasets with at least one missing categorical feature described in Tables C.1 and C.3 in Appendix C.2. Real signals $Y$ are available for 34 out of the 41 datasets. For training predictive models, missing numerical features are mean-imputed and missing categorical are either encoded as their own category or mode-imputed. The downstream predictive model is chosen among a regularized linear, a tree, a random forest, a two-layer neural network, and an XGBoost model (using 5-fold cross-validation on the training data).

**Results.** To quantitatively assess the effect of mode imputation, we compare the out-of-sample accuracy (measured in $R^2$ or $AUC$) with mode imputation with that obtained when encoding missingness as its own category, using a paired $t$-test (difference in means) and paired Wilcoxon test (difference in pseudo-medians). Results are reported in Table 1. We observe that mode imputation has a significant and negative (detrimental) effect on predictive power on instances with synthetic signals (for both synthetic and real design matrix and across missingness mechanisms), hence corroborating the insights from Theorem 2.1. In terms of magnitude, the average reduction in $R^2/AUC$ mostly occurs at the second decimal. Yet, we do not observe a significant effect when we predict the real signal $Y$, which suggests that other factors beyond missing data might impact the validity of Theorem 2.1 in practice (e.g., finite amount of samples, limited class of predictors).

Table 1: Difference in means and in pseudo-medians (with two-sided $p$-values) in out-of-sample accuracy from a $t$ and Wilcoxon test applied to assess the impact of mode imputation on downstream accuracy. A negative value means that mode imputation reduces accuracy.

| Design matrix $\boldsymbol{X}$ | Signal $Y$ | Missingness | # comp. | $\Delta$ mean ($p$-value) | $\Delta$ pseudo-median ($p$-value) |
|---|---|---|---|---|---|
| Syn. | Syn. - Linear | MCAR | 3,920 | -0.0758 (***) | -0.0762 (***) |
| | | Censoring | 3,920 | -0.1270 (***) | -0.1248 (***) |
| | Syn. - NN | MCAR | 3,920 | -0.0697 (***) | -0.0722 (***) |
| | | Censoring | 3,920 | -0.1214 (***) | -0.1153 (***) |
| Real | Syn. - Linear | MAR | 1,760 | -0.0303 (**) | -0.0079 (***) |
| | | NMAR | 1,760 | -0.0277 (**) | -0.0083 (***) |
| | | AM | 1,760 | -0.0303 (**) | -0.0071 (***) |
| | Syn. - NN | MAR | 1,760 | -0.0320 (**) | -0.0057 (***) |
| | | NMAR | 1,760 | -0.0357 (**) | -0.0070 (***) |
| | | AM | 1,760 | -0.0321 (***) | -0.0064 (***) |
| Real | Real | Real | 340 | 0.0073 (0.29) | -0.001 (0.19) |

*Note: p-values ***:$< 10^{-20}$; **:$< 10^{-10}$; *:$< 10^{-5}$;*

### 3.3 Continuous variables: mean imputation is consistent

**Theory.** For continuous features, one of the simplest and most widely used imputation rule is mean imputation, namely using $\mu = \mathbb{E}[X_1 | M_1 = 0]$. Further assume that $X_1 | \boldsymbol{X}_{2:d} = \boldsymbol{x}_{2:d}, M_1 = 0$ is continuous (i.e., if the 'observed' $X_1$ is continuous, conditioned on the other covariates), then, conditioned on $\boldsymbol{X}_{2:d} = \boldsymbol{x}_{2:d}$ and $M_1 = 0$, the probability that $X_1$ takes any specific value is 0 so $p_\mu(\boldsymbol{x}) = 0$ and $\alpha(\boldsymbol{x}) = 1$. Consequently, Corollary 2.2 guarantees that mean imputation-then-regress is asymptotically consistent, as already proved by Josse et al. (2019, Theorem 4). Intuitively, systematically imputing $\mu$ for $X_1$ creates a discontinuity in the distribution of $X_1^\mu$ and the events $\{X_1^\mu = \mu\}$ and $\{M_1 = 1\}$ are equal almost surely. A universally consistent downstream predictive model is then able to learn this pattern and view $X_1^\mu = \mu$ as an encoding for missingness. Observe that any constant imputation rule (e.g., 0-imputation or out-of-range imputation) satisfies this property, because it creates a mass in an otherwise-continuous distribution.

In practice, this result suggests that sophisticated imputation methods for continuous variables are not needed, and may indeed be counter-productive. We now empirically evaluate the validity of this finding.

**Experimental setting.** Our analysis comprises the same synthetic, semi-real, and real datasets. For the semi-real and real instances, we consider the 52 datasets with at least one missing numerical feature described in Tables C.3 and C.2 in Appendix C.2. Real signals $Y$ are available for 36 out of the 52 datasets. Missing categorical features are encoded as their own category and missing numerical features are either mean-imputed or imputed using the complex iterative method `mice` (van Buuren & Groothuis-Oudshoorn, 2010). The downstream predictive model is chosen among a regularized linear, a tree, a random forest, a two-layer neural network, and an XGBoost model (using 5-fold cross-validation on the training data).

For mean-imputation, we compute the empirical mean on the training data and use it to impute the missing values on both the training and testing data, in order to have the same imputation rule for the training and test set. For `mice`, we first impute the training set alone, and then impute the test set with the *imputed* training data. We discuss alternative implementations in Appendix D.

**Results.** Table 2 reports the output from paired $t$ and Wilcoxon tests to compare out-of-sample accuracy obtained when using mean-impute vs. `mice` (a negative value indicates that `mice` performs worse). We primarily comment on the differences in average accuracy ($t$-test), the difference in median accuracy supporting similar conclusions.

Table 2: Difference in means and in pseudo-medians (with two-sided $p$-values) in out-of-sample accuracy from a $t$ and Wilcoxon test applied to assess the impact of `mice` imputation on downstream accuracy. A negative value means that `mice` reduces accuracy compared with mean impute.

| Design matrix $X$ | Signal $Y$ | Missingness | # comp. | $\Delta$ mean ($p$-value) | $\Delta$ pseudo-median ($p$-value) |
|---|---|---|---|---|---|
| Syn. | Syn. - Linear | MCAR | 3,920 | 0.0043 (0.001) | 0.0112 (***) |
| | | Censoring | 3,920 | -0.1047 (***) | -0.0908 (***) |
| | Syn. - NN | MCAR | 3,920 | 0.0318 (***) | 0.0339 (***) |
| | | Censoring | 3,920 | -0.1275 (***) | -0.1193 (***) |
| Real | Syn. - Linear | MAR | 2,180 | -0.0256 (*) | -0.0115 (***) |
| | | NMAR | 2,390 | -0.1890 (**) | -0.0106 (***) |
| | | AM | 2,390 | -0.0020 (0.31) | -0.0029 (**) |
| | Syn. - NN | MAR | 2,390 | -0.0168 (*) | -0.0088 (***) |
| | | NMAR | 2,390 | -0.0196 (**) | -0.0134 (***) |
| | | AM | 2,390 | -0.0110 (*) | -0.0069 (***) |
| Real | Real | Real | 360 | -0.0043 (0.38) | -0.0028 (0.008) |

*Note: $p$-values ***:$< 10^{-20}$; **:$< 10^{-10}$; *:$< 10^{-5}$;*

On the fully synthetic datasets, `mice` imputation leads to more accurate predictions when the data is MCAR, but is less accurate than mean impute when the data is censored. Both observations are highly statistically significant. To elicit the mechanisms at play, we report in Figure 1 the average out-of-sample $R^2$ obtained by each method as a function of the fraction of missing entries in the data (which is one of the parameters we control), for the four design settings. Across all settings, mice imputation is preferable when the proportion of missing entries is low and mean-impute-then-regress is stronger when the proportion of missing entries is larger. The only difference between the MCAR and Censoring settings is the value of the break-even point that makes both approaches comparable. This behavior can be explained by the fact that the proportion of missing entries is directly related to the number of observations available to calibrate the imputation model: As the fraction of missing entries increases, there are fewer observations available to learn how to impute so complex models like `mice` are likely to overfit and perform poorly. However, we do not observe such a clear pattern for the impact of the sample size directly (Figure E.3).

On the semi-real and real instances, we observe that the impact of `mice`-then-regress is negative and significant ($p$-value $< 10^{-5}$) in five cases out of 7, and not statistically significant in the remaining 2 cases. These mixed results are consistent with our theoretical findings: According to Corollary 2.2, both mean impute and `mice` could be asymptotically consistent so there is no reason to expect one to be systematically better than the other. Nonetheless, the strong performance of mean-impute, despite its simplicity, is remarkable. Yet, we should note that the magnitude of the effect in average accuracy is smaller than for mode imputation in the previous section.

## 4 Discussion

We now discuss the implications and limitations of our findings. We first discuss how our analysis calls into question the relevance of the MAR assumption for prediction. We then highlight two limitations of our approach: the infinite-data assumption in Theorem 2.1, and the discrepancy in numerical results between real and synthetic data. We propose simple ways to alleviate these limitations, and identify avenues for further research on these topics.

**Relevance of the MAR assumption.** A striking feature of the theoretical consistency guarantee for impute-then-regress pipelines (Theorem 2.1) is the absence of the MAR assumption (the same observation holds for Theorem 3.1 in Le Morvan et al., 2021). Empirically, however, we observe that the relative performance of `mice` vs. mean imputation on synthetic data depends strongly on the missingness mechanisms.

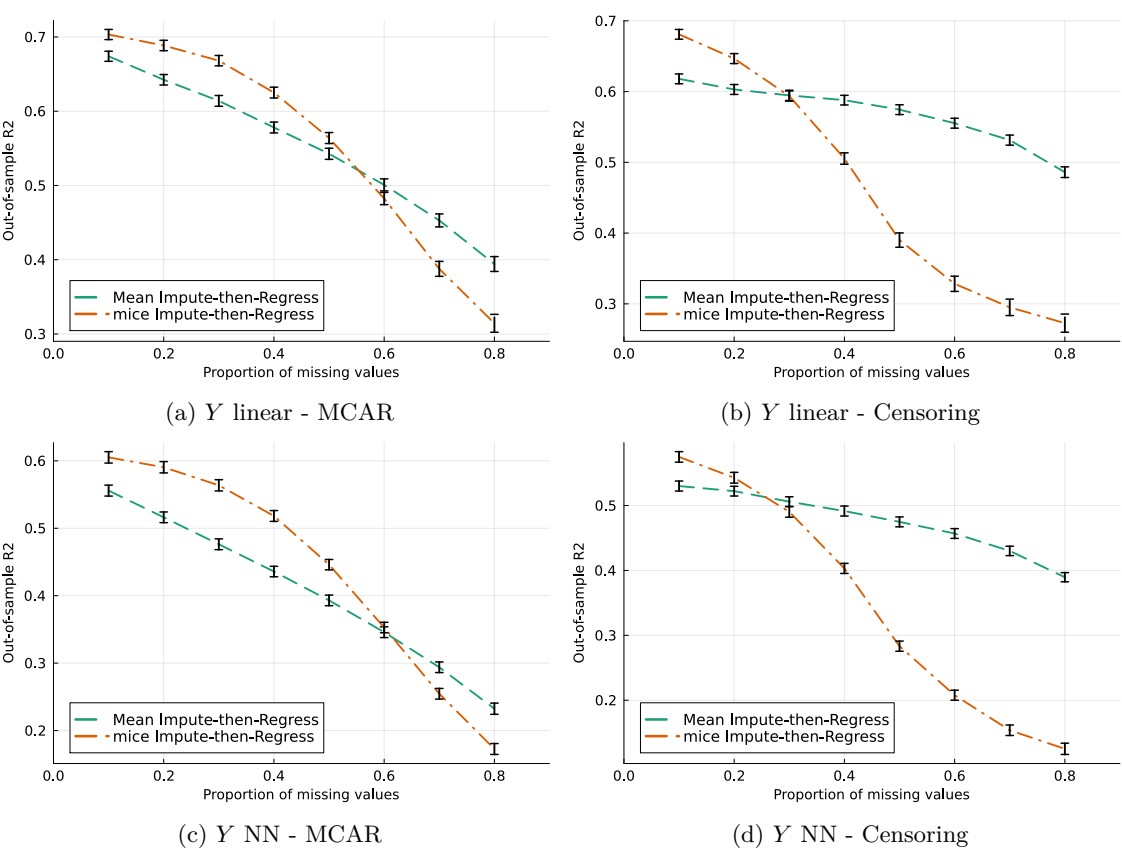

(a) $Y$ linear - MCAR

(b) $Y$ linear - Censoring

(c) $Y$ NN - MCAR

(d) $Y$ NN - Censoring

Figure 1: Average out-of-sample $R^2$ of `mice`-then-regress and mean-impute-then-regress on fully synthetic data, as the proportion of missing entries increases. Results are averaged over 50 different sample size and 10 training/test splits.

Thus, one might wonder what the true role of the missingness mechanism is. Given a generative model for $(\boldsymbol{X}, Y)$, the missingness mechanisms impact the shape and regularity of the function $\mathbb{E}[Y|\boldsymbol{o}(\boldsymbol{X}, \boldsymbol{M}), \boldsymbol{M}]$, hence its learnability. Since Theorem 2.1 considers a universally consistent predictive model, we can always guarantee consistency with infinite data. For a finite amount of data, however, the shape and regularity of the function—and so, indirectly, the missingness mechanism—will impact the performance of a method. Based on this observation, one might wonder whether alternative assumptions would be more relevant to study and develop methods for prediction with missing data.

In particular, influenced by the insights from inference, practitioners tend to consider MAR as a favorable situation compared with NMAR. Indeed, textbooks in the field, such as Hastie et al. (2001, Chapter 9.6) or Kuhn et al. (2013, Chapter 3.4), tend to present MAR favorably. However, one should emphasize that, as far as prediction is concerned, NMAR can be a blessing and not a curse, because missingness can sometimes be used as a powerful predictor of the outcome of interest (i.e., predictive missingness). We formalize this observation in the setting of Theorem 2.1 (proof in Appendix B):

**Theorem 4.1.** *Consider two missingness mechanisms $M_1$ and $M_1'$ leading to the same proportion of missing entries $\mathbb{P}(M_1 = 1) = \mathbb{P}(M_1' = 1) = p$. Further assume that, conditioned on $\boldsymbol{X}_{2:d}$, $M_1'$ is independent of $Y$ and $X_1$ (MAR). Then the optimal prediction rule achieves a lower prediction error under $M_1$ than under $M_1'$ if and only if the following condition holds:*

$$(1-p)^2 \psi_0\left(\boldsymbol{X}, \boldsymbol{X}\right) + p^2 \psi_1(\boldsymbol{X}_{2:d}, \boldsymbol{X}_{2:d}) + p(1-p)\psi_0(\boldsymbol{X}_{2:d}, \boldsymbol{X}) - p(1-p)\psi_1(\boldsymbol{X}, \boldsymbol{X}_{2:d}) \geq 0,$$

*where*

$$\psi_i(A, B) = \mathbb{E}\left[\left(\mathbb{E}\left[Y|A\right] - \mathbb{E}\left[Y|B, M_1\right]\right)^2 |M_1 = i\right] \geq 0.$$

We now provide simple examples of each situation.

*Example* 4.2 (Prediction benefits from NMAR). Let $X_1$ be a Bernoulli random variable with parameter $1/2$, and $Y = X_1$. We compare two missingness patterns: $M_1 = X_1$, and $M_1' \perp\!\!\!\perp X_1$, another Bernoulli random variable with parameter $1/2$. Then, the optimal predictor under $M_1$ is $Y = X_1$ when $M_1 = 0$ and $Y = M_1 = 1$ when $M_1 = 1$, with empirical risk $R = 0$. Meanwhile, the Bayes-optimal predictor under $M_1'$ is $Y = X_1$ when $M_1' = 0$ and $Y = 1/2$ when $M_1' = 1$, with empirical risk $R' = 1/8$.

*Example* 4.3 (Prediction benefits from MAR). Let $d = 2$, and let $X_2$, $U$, $V$ be three independent Bernoulli random variables with parameter $1/2$. Define $X_1 = X_2\mathbb{1}(U = 0) + V\mathbb{1}(U = 1)$ and let $Y = X_1$. We compare two missingness patterns: $M_1 = U$, and $M_1'$ an independent Bernoulli random variable with parameter $1/2$. Then the Bayes-optimal learner under $M_1$ is $X_1$ when $M_1 = 0$ and $1/2$ when $M_1 = 1$, with empirical risk $R = 1/8$. In contrast, the Bayes-optimal learner under $M_1'$ is $X_1$ when $M_1' = 0$, and $X_2/2 + 1/4$ when $M_1' = 1$, with empirical risk $R' = 3/32$.

Such a result highlights the fact that the MAR/NMAR distinction does not correctly classify missing data into "good"/"bad" cases and that a better taxonomy is needed. This insight is valuable because missing data mechanisms are inherently unobservable and because the MAR assumption is obviously violated in many industrial applications, e.g., in pricing and revenue management (Alles et al., 2000; Wang, 2022; Pauphilet, 2022).

**Finite-sample and parametric models.** One limitation of Theorem 2.1 is that it relies on two restrictive assumptions, namely that the number of observations grows to infinity and that the downstream predictive model is universally consistent. Universally consistent models, however, require large datasets, while real-world data is often limited. Hence, in practice, one might favor parametric classes of predictors (instead of universally consistent learning algorithms), for which Theorem 2.1 does not apply. More work is needed to design tailored classes of parametric models for prediction with missing data (such as Le Morvan et al., 2020a) or to understand the finite-sample performance of impute-then-regress pipelines.

In the latter case, we believe that the interactions between the model complexity of the imputation rule and that of the downstream regression are not yet well-understood and provide exciting grounds for future research. Figure 2 illustrates this phenomenon by representing the out-of-sample $R^2$ of four impute-then-regress pipelines (using `mice`/mean imputation and a linear/random forest regressor), as the number of

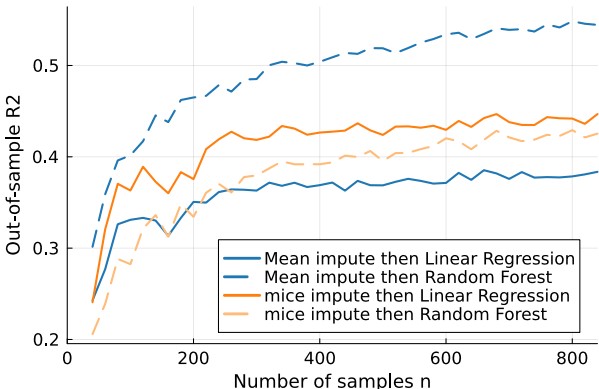

Figure 2: Out-of-sample $R^2$ on `mice`-regress and mean-impute-then-regress on synthetic data with non-linear signal, NMAR data, and 40% of missing entries, as the number of samples $n$ increases. We report the performance of two different downstream predictors: Linear (LASSO) regression and random forest. Results are averaged over 10 training/test splits.

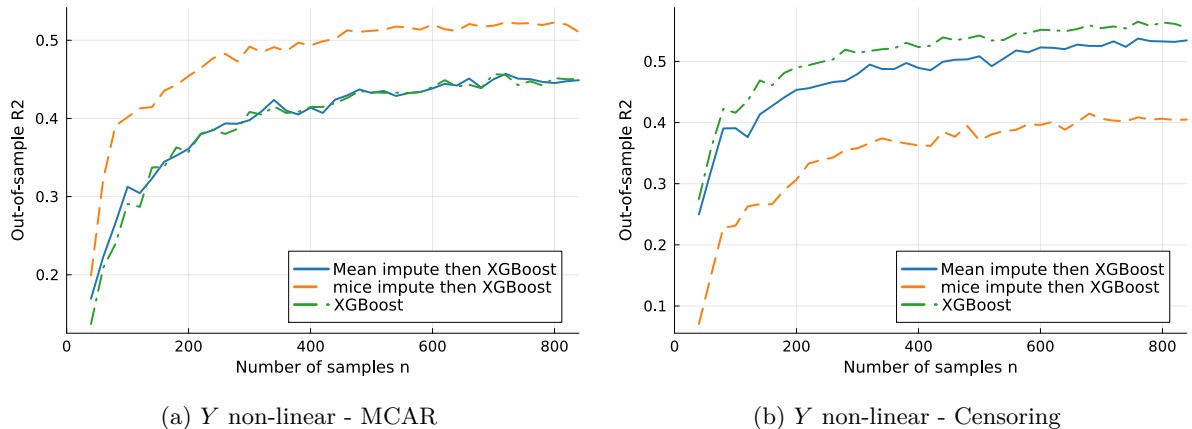

(a) $Y$ non-linear - MCAR

(b) $Y$ non-linear - Censoring

Figure 3: Average out-of-sample $R^2$ of XGBoost with mean impute, `mice`, or no imputation method, on synthetic data with non-linear signal, NMAR missing data, and 40% of missing entries, as the number of samples $n$ increases. Results are averaged over 10 training/test splits.

observations increases, for our synthetically generated datasets, with censored data and non-linear synthetic signal $Y$. When a linear regression model is used, we find that `mice` consistently outperforms mean impute (across all values of $n$) and is strictly more accurate asymptotically. When using a random forest regressor, however, we observe the exact opposite. These empirical observations complement some theoretical findings from Le Morvan et al. (Section 4 2021) and demonstrate the complexity of the inter-dependencies between the classes of models used for imputation and regression.

This observation holds also for models that can handle missing data natively such as tree-based methods. For XGBoost, for example, we compare how the out-of-sample accuracy increases with the number $n$ of training samples, for mean impute followed by XGBoost, `mice` impute followed by XGBoost, and XGBoost applied on missing data directly in Figure 3. We observe that the relative performance of the no-imputation strategy (especially compared with an imputation method like `mice`) varies drastically with the missingness pattern. We can make a similar observation for random forest (Figure E.4). These observations highlight that, although 'hassle-free', methods which can handle missing data directly come at a cost in terms of predictive power.

**Realistic synthetic data.** A second limitation is that, while our empirical experiments largely confirm the theoretical findings when synthetic data is used, the conclusions on real data (and to a lesser extent on semi-real data) are noticeably less obvious. This observation raises the question of the validity of analysis conducted with synthetic data in the literature, especially for research related with missing data. Algorithms for missing data imputation are almost always validated on synthetically generated data or at least synthetically amputed data, in settings where the gains (both in imputation error and downstream accuracy) can be significant (e.g., Bertsimas et al., 2018; 2021). On the contrary, we observe on real-world design matrices and missingness patterns (both for synthetic and real signals $Y$) that an imputation method as sophisticated as `mice` leads, on average, to less accurate predictions than simple mean imputation. Synthetic data are useful to researchers because they often allow for more extensive numerical validation — but this is only true if they accurately represent how data go missing in practice. Our findings thus highlight the need for more realistic and accessible models for missingness to allow researchers to benchmark and develop models that are better suited to real-world data with missing entries.

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

# A  Proof of Theorem 2.1

*Proof.* Since the downstream predictive model is trained after $\mu$-imputation, the learner is trained not on $(\boldsymbol{o}(\boldsymbol{X}, M_1), M_1)$ but on its imputed version $\boldsymbol{X}^\mu$. Accordingly, asymptotically, $f_{\mu\text{-impute}}(\boldsymbol{x}) = \mathbb{E}[Y|\boldsymbol{X}^\mu = \boldsymbol{x}]$. We now relate the conditional expectation of $Y|X_1^\mu = x_1^\mu, \boldsymbol{X}_{2:d} = \boldsymbol{x}_{2:d}$ with those of $Y|M_1 = 1, \boldsymbol{X}_{2:d} = \boldsymbol{x}_{2:d}$ and $Y|M_1 = 0, X_1 = x_1^\mu, \boldsymbol{X}_{2:d} = \boldsymbol{x}_{2:d}$.

For random variables $Z_1, Z_2$, we denote $g_{Z_1}(z_1)dz_1$ the distribution function of $Z_1$ and $g_{Z_1}(z_1|Z_2 = z_2)dz_1$ the distribution function of $Z_1|Z_2 = z_2$.

We condition on the event $\{\boldsymbol{X}_{2:d} = \boldsymbol{x}_{2:d}\}$ for some $\boldsymbol{x}_{2:d}$ such that $g_{\boldsymbol{X}_{2:d}}(\boldsymbol{x}_{2:d}) > 0$. All our reasoning is conditioned on $\{\boldsymbol{X}_{2:d} = \boldsymbol{x}_{2:d}\}$, e.g., all distributions/probabilities are conditional distributions/probabilities, but we omit the dependency on $\boldsymbol{x}_{2:d}$ for simplicity. We also denote $\mu := \mu(\boldsymbol{x}_{2:d})$.

The joint density for $(X_1, M_1, X_1^\mu)$ is $\left(\mathbf{1}_{m_1=1}\mathbf{1}_{x_1^\mu=\mu} + \mathbf{1}_{m_1=0}\mathbf{1}_{x_1^\mu=x_1}\right) g_{(X_1,M_1)}(x_1, m_1)dx_1^\mu dx_1 dm_1$. By integrating over $(X_1, M_1)$, we obtain the density of $X_1^\mu$:

$$h(x_1^\mu)dx_1^\mu := \mathbb{P}(M_1 = 1)\mathbf{1}_{x_1^\mu=\mu}dx_1^\mu + g_{(X_1,M_1)}(x_1^\mu, 0)dx_1^\mu.$$

Note that

$$h(x_1^\mu)dx_1^\mu = \begin{cases} g_{(X_1,M_1)}(x_1^\mu, 0)dx_1^\mu & \text{if } x_1^\mu \neq \mu, \\ \mathbb{P}(M_1 = 1)dx_1^\mu + g_{(X_1,M_1)}(\mu, 0)dx_1^\mu & \text{if } x_1^\mu = \mu. \end{cases}$$

Furthermore, let us denote $h(y, x_1^\mu)$ the joint density of $(Y, X_1^\mu)$. Since the joint density for $(Y, X_1, M_1, X_1^\mu)$ is $g_Y(y|X_1 = x_1, M_1 = m_1)\left(\mathbf{1}_{m_1=1}\mathbf{1}_{x_1^\mu=\mu} + \mathbf{1}_{m_1=0}\mathbf{1}_{x_1^\mu=x_1}\right) g_{(X_1,M_1)}(x_1, m_1)dy dx_1^\mu dx_1 dm_1$, we have

$$h(y, x_1^\mu)dy dx_1^\mu = \mathbf{1}_{x_1^\mu=\mu}g_Y(y|M_1 = 1)\mathbb{P}(M_1 = 1)dy dx_1^\mu + g_Y(y|X_1 = x_1^\mu, M_1 = 0)g_{(X_1,M_1)}(x_1^\mu, 0)dy dx_1^\mu.$$

Accordingly,

$$\int yh(y, x_1^\mu)dy = \mathbf{1}_{x_1^\mu=\mu}\mathbb{P}(M_1 = 1)\int yg_Y(y|M_1 = 1)dy + g_{(X_1,M_1)}(x_1^\mu, 0)\int yg_Y(y|X_1 = x_1^\mu, M_1 = 0)dy$$

$$= \mathbf{1}_{x_1^\mu=\mu}\mathbb{P}(M_1 = 1)\mathbb{E}[Y|M_1 = 1] + g_{(X_1,M_1)}(x_1^\mu, 0)\mathbb{E}[Y|M_1 = 0, X_1 = x_1^\mu].$$

All together, distinguishing the case where $(X_1, M_1)$ is continuous/discrete and denoting

$$\alpha := \frac{\mathbb{P}(M_1 = 1)}{\mathbb{P}(M_1 = 1) + \mathbb{P}(X_1 = \mu, M_1 = 0)},$$

we obtain

$$\mathbb{E}[Y|X_1^\mu = x_1^\mu] = \begin{cases} \mathbb{E}[Y|X_1 = x_1^\mu, M_1 = 0] & \text{if } x_1^\mu \neq \mu, \\ \alpha\,\mathbb{E}[Y|M_1 = 1] + (1 - \alpha)\,\mathbb{E}[Y|X_1 = \mu, M_1 = 0] & \text{if } x_1^\mu = \mu. \end{cases}$$

$\square$

# B  Proof of Theorem 4.1 and Examples

The quantities $\psi_i(A, B)$ in Theorem 4.1 measure the distance between $\mathbb{E}[Y|A]$ and $\mathbb{E}[Y|B, M_1]$ conditional on $M_1 = i$. In other words, these quantities compare the predictive power (on $Y$) of feature set $A$ and feature set $\{B, M_1\}$. We note from the signs in the condition in Theorem 4.1 that a sufficient condition for the NMAR setting to yield higher predictive power is $\mathbb{E}\left[(\mathbb{E}[Y|\boldsymbol{X}] - \mathbb{E}[Y|\boldsymbol{X}_{2:d}, M_1])^2 | M_1 = i\right] = 0$, corresponding to the case where $M_1$ is a perfect substitute for $X_1$. This condition is not necessary since the first three terms can be strictly positive: in this case, NMAR can lead to higher predictive power as long as replacing the value $X_1$ with the fact that it is missing preserves enough information about the value of $Y$.

*Proof.* Proof of Theorem 4.1 Let $R$ ($R'$) designate the optimal empirical risk under missingness $M_1$ ($M_1'$).

$$R = (1-p)\mathbb{E}\left[(Y - \mathbb{E}\left[Y|\boldsymbol{X}, M_1\right])^2|M_1 = 0\right] + p\mathbb{E}\left[(Y - \mathbb{E}\left[Y|\boldsymbol{X}_{2:d}, M_1\right])^2|M_1 = 1\right],$$
$$R' = (1-p)\mathbb{E}\left[(Y - \mathbb{E}\left[Y|\boldsymbol{X}, M_1'\right])^2|M_1' = 0\right] + p\mathbb{E}\left[(Y - \mathbb{E}\left[Y|\boldsymbol{X}_{2:d}, M_1'\right])^2|M_1' = 1\right].$$

Because $M_1'$ is independent of $Y$ given $\boldsymbol{X}_{2:d}$, we can write:

$$R' = (1-p)\mathbb{E}\left[(Y - \mathbb{E}\left[Y|\boldsymbol{X}\right])^2|M_1' = 0\right] + p\mathbb{E}\left[(Y - \mathbb{E}\left[Y|\boldsymbol{X}_{2:d}\right])^2|M_1' = 1\right]$$
$$= (1-p)\mathbb{E}\left[(Y - \mathbb{E}\left[Y|\boldsymbol{X}\right])^2\right] + p\mathbb{E}\left[(Y - \mathbb{E}\left[Y|\boldsymbol{X}_{2:d}\right])^2\right],$$

where the last equality follows from the MAR assumption. Then we can apply the tower rule, conditioning on $M_1$:

$$R' = (1-p)\mathbb{E}\left[\mathbb{E}\left[(Y - \mathbb{E}\left[Y|\boldsymbol{X}\right])^2|M_1\right]\right] + p\mathbb{E}\left[\mathbb{E}\left[(Y - \mathbb{E}\left[Y|\boldsymbol{X}_{2:d}\right])^2|M_1\right]\right]$$
$$= (1-p)^2\mathbb{E}\left[(Y - \mathbb{E}\left[Y|\boldsymbol{X}\right])^2|M_1 = 0\right] + p(1-p)\mathbb{E}\left[(Y - \mathbb{E}\left[Y|\boldsymbol{X}\right])^2|M_1 = 1\right]$$
$$+ p(1-p)\mathbb{E}\left[(Y - \mathbb{E}\left[Y|\boldsymbol{X}_{2:d}\right])^2|M_1 = 0\right] + p^2\mathbb{E}\left[(Y - \mathbb{E}\left[Y|\boldsymbol{X}_{2:d}\right])^2|M_1 = 1\right]$$
$$=: (1-p)^2 A + p(1-p)(B + C) + p^2 D.$$

Then, we can modify the terms $A$, $B$, $C$, and $D$ as follows:

$$A = \mathbb{E}\left[(Y - \mathbb{E}\left[Y|\boldsymbol{X}\right])^2|M_1 = 0\right]$$
$$= \mathbb{E}\left[(Y - \mathbb{E}\left[Y|\boldsymbol{X}, M_1\right] + \mathbb{E}\left[Y|\boldsymbol{X}, M_1\right] - \mathbb{E}\left[Y|\boldsymbol{X}\right])^2|M_1 = 0\right]$$
$$= \mathbb{E}\left[(Y - \mathbb{E}\left[Y|\boldsymbol{X}, M_1\right])^2|M_1 = 0\right] + \mathbb{E}\left[(\mathbb{E}\left[Y|\boldsymbol{X}, M_1\right] - \mathbb{E}\left[Y|\boldsymbol{X}\right])^2|M_1 = 0\right]$$
$$+ 2\mathbb{E}\left[(Y - \mathbb{E}\left[Y|\boldsymbol{X}, M_1\right])(\mathbb{E}\left[Y|\boldsymbol{X}, M_1\right] - \mathbb{E}\left[Y|\boldsymbol{X}\right])|M_1 = 0\right]$$
$$= \mathbb{E}\left[(Y - \mathbb{E}\left[Y|\boldsymbol{X}, M_1\right])^2|M_1 = 0\right] + \psi_0(\boldsymbol{X}, \boldsymbol{X}),$$

since $\mathbb{E}\left[Y|\boldsymbol{X}, M_1\right] - \mathbb{E}\left[Y|\boldsymbol{X}\right]$ is constant when conditioning on $\boldsymbol{X}$ and $M_1$, and $\mathbb{E}\left[(Y - \mathbb{E}\left[Y|\boldsymbol{X}, M_1\right])|\boldsymbol{X}, M_1\right] = 0$. Similarly, we can show that

$$B = \mathbb{E}\left[(Y - \mathbb{E}\left[Y|\boldsymbol{X}_{2:d}, M_1\right])^2|M_1 = 1\right] - \psi_1(\boldsymbol{X}, \boldsymbol{X}_{2:d}),$$
$$C = \mathbb{E}\left[(Y - \mathbb{E}\left[Y|\boldsymbol{X}, M_1\right])^2|M_1 = 0\right] + \psi_0(\boldsymbol{X}_{2:d}, \boldsymbol{X}),$$
$$D = \mathbb{E}\left[(Y - \mathbb{E}\left[Y|\boldsymbol{X}_{2:d}, M_1\right])^2|M_1 = 1\right] + \psi_1(\boldsymbol{X}_{2:d}, \boldsymbol{X}_{2:d}).$$

Putting it all together yields:

$$R' = (1-p)\mathbb{E}\left[(Y - \mathbb{E}\left[Y|\boldsymbol{X}, M_1\right])^2|M_1 = 0\right] + p\mathbb{E}\left[(Y - \mathbb{E}\left[Y|\boldsymbol{X}_{2:d}, M_1\right])^2|M_1 = 1\right]$$
$$+ (1-p)^2\psi_0(\boldsymbol{X}, \boldsymbol{X}) + p^2\psi_1(\boldsymbol{X}_{2:d}, \boldsymbol{X}_{2:d}) + p(1-p)\psi_0(\boldsymbol{X}_{2:d}, \boldsymbol{X}) - p(1-p)\psi_1(\boldsymbol{X}, \boldsymbol{X}_{2:d}).$$

Recognizing that the first two terms in the above expression are equal to $R$ completes the proof. $\qquad\square$

## C  Description of the Data and Evaluation Methodology

In this section, we describe the datasets we used in our numerical experiments, as well as various implementation details. In line with other works in the literature, we conduct some of our experiments on synthetic data, where we have full control over the design matrix $\boldsymbol{X}$, the missingness pattern $\boldsymbol{M}$, and the signal $Y$. We also contrast the results obtained on these synthetic instances with real world instances from the UCI Machine Learning Repository and the RDatasets Repository[2].

Note that all experiments were performed on a Intel Xeon E5—2690 v4 2.6GHz CPU core using 8 GB RAM.

---

[2]https://archive.ics.uci.edu and https://github.com/vincentarelbundock/Rdatasets

### C.1 Synthetic data generation

As in Le Morvan et al. (2020b, Section 7), we generate a multivariate vector $\boldsymbol{X}$ from a multivariate Gaussian with mean 0 and covariance matrix $\boldsymbol{\Sigma} := \boldsymbol{B}\boldsymbol{B}^\top + \epsilon\mathbb{I}$ where $\boldsymbol{B} \in \mathbb{R}^{d \times r}$ with i.i.d. standard Gaussian entries and $\epsilon > 0$ is chosen small enough so that $\boldsymbol{\Sigma} \succ \boldsymbol{0}$. We fix $d = 10$ and $r = 5$ in our experiments. For experiments requiring discrete/categorical features (Section 3.2), we convert the entries of $\boldsymbol{X}$ into $\{0, 1\}$ by sampling each entry according to $Ber(\text{logit}(X_{ij}))$. We generate a $n$ observations, $n \in \{40, 60, \dots, 1000\}$, for the training data and $5,000$ observations for the test data.

We then generate signals $Y = f(\boldsymbol{X}) + \varepsilon$, where $f(\cdot)$ is a predefined function and $\varepsilon$ is a centered normal random noise whose variance is calibrated to achieve a target signal-to-noise ratio $SNR$. We choose $SNR = 2$ in our experiments. We use functions $f(\cdot)$ of the following forms:

- **Linear model:** $f(\boldsymbol{x}) = b + \boldsymbol{w}^\top \boldsymbol{x}$ where $b \sim \mathcal{N}(0, 1)$ and $w_j \sim \mathcal{U}([-1, 1])$.

- **Neural Network (NN) model:** $f(\boldsymbol{x})$ corresponds to the output function of a 2-layer neural network with 10 hidden nodes, ReLU activation functions, and random weights and intercept for each node.

We compute $f(\boldsymbol{x})$ using a random subset of $k$ out of the $d$ features only, with $k = 5$.

Finally, for a given fraction of missing entries $p$, we generate missing entries according to mechanisms

- **Missing Completely At Random (MCAR):** For each observation and for each feature $j \in \{1, \dots, d\}$, we sample $M_j \sim Bern(p)$ independently (for each feature and each observation).

- **Not Missing At Random (NMAR) - Censoring:** We set $M_j = 1$ whenever the value of $X_j$ is above the $(1-p)$th percentile.

For the fraction of missing entries, we consider the different values $p \in \{0.1, 0.2, \dots, 0.8\}$.

With this methodology, we generate a total of 49 training sets, with 2 categories of signal $Y$, 2 missingness mechanisms, and 8 proportion of missing entries, i.e., $1,568$ different instances.

We measure the predictive power of a method in terms of average out-of-sample $R^2$. We use $R^2$, which is a scaled version of the mean squared error, to allow for a fair comparison and aggregation of the results across datasets and generative models.

### C.2 Real-world design matrix

In addition to synthetic data, we also assemble a corpus of 63 publicly available datasets with missing data, from the UCI Machine Learning Repository and the RDatasets Repository. Tables C.1, C.2, and C.3 present summary statistics for the datasets with only categorical features missing, only continuous features missing, and both categorical and continuous features missing respectively. We use the datasets presented in Tables C.1 and C.3 in Section 3.2, to empirically validate that, for categorical features, missingness should be encoded as a category instead of using mode imputation. Then, we use the datasets from Tables C.2 and C.3 in Section 3.3, to compare the performance of different impute-then-regress strategies and our adaptive regression models.

For these datasets, we consider two categories of signal $Y$, real-world and synthetic signals.

| Dataset | $n$ | #features | #missing cont. | #missing cat. | $|\mathcal{M}|$ | $d$ | $Y$ |
|---|---|---|---|---|---|---|---|
| Ecdat-Males | 4360 | 37 | 0 | 4 | 2 | 38 | cont. |
| mushroom | 8124 | 116 | 0 | 4 | 2 | 117 | bin. |
| post-operative-patient | 90 | 23 | 0 | 4 | 2 | 24 | bin. |
| breast-cancer | 286 | 41 | 0 | 7 | 3 | 43 | bin. |
| heart-disease-cleveland | 303 | 28 | 0 | 8 | 3 | 30 | bin. |
| COUNT-loomis | 384 | 9 | 0 | 9 | 4 | 12 | cont. |
| Zelig-coalition2 | 314 | 24 | 0 | 14 | 2 | 25 | NA |
| shuttle-landing-control | 15 | 16 | 0 | 16 | 6 | 21 | bin. |
| congressional-voting-records | 435 | 32 | 0 | 32 | 60 | 48 | bin. |
| lung-cancer | 32 | 157 | 0 | 33 | 3 | 159 | bin. |
| soybean-large | 307 | 98 | 0 | 98 | 8 | 132 | bin. |

Table C.1: Description of the 11 datasets in our library where the features affected by missingness are categorical features only. $n$ denotes the number of observations. The columns '#features', '#missing cont.', and '#missing cat.' report the total number of features, the number of continuous features affected by missingness, and the number of categorical features affected by missingness, respectively. $|\mathcal{M}|$ correspond to the number of unique missingness patterns $\boldsymbol{m} \in \{0,1\}^d$ observed, where $d$ is the total number of features after one-hot-encoding of the categorical features. The final column $Y$ indicates whether the dependent variable is binary or continuous (if available).

| Dataset | $n$ | #features | #missing cont. | #missing cat. | $|\mathcal{M}|$ | $d$ | $Y$ |
|---|---|---|---|---|---|---|---|
| auto-mpg | 398 | 13 | 1 | 0 | 2 | 13 | cont. |
| breast-cancer-wisconsin-original | 699 | 9 | 1 | 0 | 2 | 9 | bin. |
| breast-cancer-wisconsin-prognostic | 198 | 32 | 1 | 0 | 2 | 32 | bin. |
| dermatology | 366 | 130 | 1 | 0 | 2 | 130 | cont. |
| ggplot2-movies | 58788 | 34 | 1 | 0 | 2 | 34 | NA |
| indian-liver-patient | 583 | 11 | 1 | 0 | 2 | 11 | bin. |
| rpart-car.test.frame | 60 | 81 | 1 | 0 | 2 | 81 | bin. |
| Ecdat-MCAS | 180 | 13 | 2 | 0 | 3 | 13 | cont. |
| MASS-Cars93 | 93 | 64 | 2 | 0 | 3 | 64 | cont. |
| car-Davis | 200 | 6 | 2 | 0 | 4 | 6 | NA |
| car-Freedman | 110 | 4 | 2 | 0 | 2 | 4 | NA |
| car-Hartnagel | 37 | 8 | 2 | 0 | 2 | 8 | NA |
| datasets-airquality | 153 | 4 | 2 | 0 | 4 | 4 | NA |
| mlmRev-Gcsemv | 1905 | 77 | 2 | 0 | 3 | 77 | NA |
| MASS-Pima.tr2 | 300 | 7 | 3 | 0 | 6 | 7 | bin. |
| Ecdat-RetSchool | 3078 | 37 | 4 | 0 | 8 | 37 | cont. |
| arrhythmia | 452 | 391 | 5 | 0 | 7 | 391 | cont. |
| boot-neuro | 469 | 6 | 5 | 0 | 9 | 6 | NA |
| reshape2-french_fries | 696 | 9 | 5 | 0 | 4 | 9 | NA |
| survival-mgus | 241 | 15 | 5 | 0 | 11 | 15 | bin. |
| sem-Tests | 32 | 6 | 6 | 0 | 8 | 6 | NA |
| robustbase-ambientNOxCH | 366 | 13 | 13 | 0 | 45 | 13 | NA |

Table C.2: Description of the 22 datasets in our library where the features affected by missingness are numerical features only. $n$ denotes the number of observations. The columns '#features', '#missing cont.', and '#missing cat.' report the total number of features, the number of continuous features affected by missingness, and the number of categorical features affected by missingness, respectively. $|\mathcal{M}|$ correspond to the number of unique missingness patterns $\boldsymbol{m} \in \{0,1\}^d$ observed, where $d$ is the total number of features after one-hot-encoding of the categorical features. The final column $Y$ indicates whether the dependent variable is binary or continuous (if available).

| Dataset | $n$ | #features | #missing cont. | #missing cat. | $|\mathcal{M}|$ | $d$ | $Y$ |
|---|---|---|---|---|---|---|---|
| pscl-politicalInformation | 1800 | 1440 | 1 | 1431 | 3 | 1441 | bin. |
| car-SLID | 7425 | 8 | 2 | 3 | 8 | 9 | NA |
| rpart-stagec | 146 | 15 | 2 | 3 | 4 | 16 | NA |
| Ecdat-Schooling | 3010 | 51 | 2 | 8 | 9 | 53 | cont. |
| mammographic-mass | 961 | 15 | 2 | 13 | 9 | 18 | bin. |
| cluster-plantTraits | 136 | 68 | 2 | 37 | 16 | 85 | NA |
| mlmRev-star | 24613 | 122 | 2 | 72 | 19 | 128 | cont. |
| car-Chile | 2532 | 14 | 3 | 3 | 7 | 15 | bin. |
| heart-disease-hungarian | 294 | 25 | 3 | 17 | 16 | 31 | bin. |
| heart-disease-switzerland | 123 | 26 | 3 | 18 | 12 | 32 | bin. |
| ggplot2-msleep | 83 | 35 | 3 | 29 | 15 | 37 | NA |
| survival-cancer | 228 | 13 | 4 | 4 | 8 | 14 | cont. |
| heart-disease-va | 200 | 25 | 4 | 13 | 18 | 30 | bin. |
| MASS-survey | 237 | 24 | 4 | 19 | 8 | 29 | NA |
| hepatitis | 155 | 32 | 5 | 20 | 21 | 42 | bin. |
| automobile | 205 | 69 | 6 | 2 | 7 | 70 | cont. |
| echocardiogram | 132 | 8 | 6 | 2 | 13 | 9 | bin. |
| thyroid-disease-allbp | 2800 | 52 | 6 | 46 | 25 | 54 | bin. |
| thyroid-disease-allhyper | 2800 | 52 | 6 | 46 | 25 | 54 | bin. |
| thyroid-disease-allhypo | 2800 | 52 | 6 | 46 | 25 | 54 | bin. |
| thyroid-disease-allrep | 2800 | 52 | 6 | 46 | 25 | 54 | bin. |
| thyroid-disease-dis | 2800 | 52 | 6 | 46 | 25 | 54 | bin. |
| thyroid-disease-sick | 2800 | 52 | 6 | 46 | 25 | 54 | bin. |
| survival-pbc | 418 | 27 | 7 | 17 | 8 | 32 | bin. |
| thyroid-disease-sick-euthyroid | 3163 | 43 | 7 | 36 | 23 | 44 | bin. |
| horse-colic | 300 | 60 | 7 | 52 | 171 | 73 | bin. |
| plyr-baseball | 21699 | 296 | 9 | 14 | 18 | 297 | NA |
| communities-and-crime | 1994 | 126 | 22 | 3 | 4 | 127 | cont. |
| communities-and-crime-2 | 2215 | 129 | 22 | 3 | 4 | 130 | cont. |
| wiki4he | 913 | 73 | 44 | 24 | 236 | 78 | cont. |

Table C.3: Description of the 30 datasets in our library where the features affected by missingness are numerical and categorical. $n$ denotes the number of observations. The columns '#features', '#missing cont.', and '#missing cat.' report the total number of features, the number of continuous features affected by missingness, and the number of categorical features affected by missingness, respectively. $|\mathcal{M}|$ correspond to the number of unique missingness patterns $\boldsymbol{m} \in \{0,1\}^d$ observed, where $d$ is the total number of features after one-hot-encoding of the categorical features. The final column $Y$ indicates whether the dependent variable is binary or continuous (if available).

### C.2.1 Real signal $Y$

46 out of the 63 datasets had an identified target variable $Y$, which could be continuous or binary. If $Y$ is categorical with more than 1 category, we considered the binary one-vs-all classification task using the first (alphabetical order) category. For regression (resp. classification) tasks, we use the mean squared error (resp. logistic log-likelihood) as the training loss and measure predictive power in terms of $R^2$ (resp. $2 \times AUC - 1$). Again, we choose this measure over mean square error (resp. accuracy or $AUC$) because it is normalized between 0 and 1, and can be more safely compared and aggregated across datasets.

### C.2.2 Synthetic signal $Y$

To generate synthetic signals $Y$, we use the same three generative models as with synthetic data in Section C.1. However, this requires knowledge of the fully observed input matrix, while we only have access to observations with missing entries, $(\boldsymbol{o}(\boldsymbol{x}^{(i)}, \boldsymbol{m}^{(i)}), \boldsymbol{m}^{(i)})$, $i = 1, \ldots, n$. Therefore, we first generate a fully observed version of the data by performing missing data imputation using the R package `missForest` (Stekhoven & Bühlmann, 2012), obtaining a new dataset $\{(\boldsymbol{x}_{\text{full}}^{(i)}, \boldsymbol{m}^{(i)})\}_{i \in [n]}$. We use this dataset to generate synthetic signals $Y$, using the three types of signals described in Section C.1: linear, tree, and neural network.

Regarding the relationship between the missingness pattern $\boldsymbol{M}$ and the signal $Y$, we consider three mechanisms:

- **MAR:** In this setting, we pass $k = \min(10, d)$ coordinates of $\boldsymbol{x}_{\text{full}}$ as input to the function $f(\cdot)$. Out of these $k$ features, we explicitly control $k_{missing} \in \{0, \ldots, k\}$, the number of features contributing to the signal that are affected by missingness. Hence, in this setting, the resulting response $Y$ depends directly on the covariates $\boldsymbol{X}$ but not on the missingness pattern $\boldsymbol{M}$. However, we do not control the correlation between $\boldsymbol{X}$ and $\boldsymbol{M}$ for two reasons: First, they both come from a real-world dataset which might not satisfy the MAR assumption. Second, as previously observed (Section 2.2), imputation does induce some correlation between the imputed dataset $\boldsymbol{X}_{full}$ and $\boldsymbol{M}$.

- **NMAR:** In the second setting, in addition to $k = 10$ coordinates of $\boldsymbol{x}_{\text{full}}$, we also pass $k_{missing}$ coordinates of $\boldsymbol{m}$, so that $Y$ is now a function of both $\boldsymbol{X}$ and $\boldsymbol{M}$.

- **Adversarially Missing (AM):** The third setting generates $Y$ in the same way as the **MAR** setting. After $Y$ is generated, however, we reallocate the missingness patterns across observations so as to ensure the data is NMAR. Formally, we consider the observations $(\boldsymbol{o}(\boldsymbol{x}_{\text{full}}^{(i)}, \boldsymbol{m}^{(\sigma_i)}), \boldsymbol{m}^{(\sigma_i)}, y^{(\sigma_i)})$, $i \in [n]$, where $\boldsymbol{\sigma}$ is the permutation maximizing the total sum of missing values $\sum_{i=1}^{n} \boldsymbol{x}_{\text{full}}^{(i)\top} \boldsymbol{m}^{(\sigma_i)}$.

For each real-world dataset, this methodology generates up to $3 \times 3 \times 11 = 66$ different instances.

All together, we obtain four experimental settings, with both synthetic and real signals $Y$. They differ in the relationships between the missingness pattern $\boldsymbol{M}$, the design matrix $\boldsymbol{X}$ and the signal $Y$ as summarized on Figure C.1.

## C.3 Evaluation pipeline

In our numerical experiments, we compare a series of impute-then-regress methods where the imputation step is performed either via mode/mean imputation or using the chained equation method `mice` (van Buuren & Groothuis-Oordshoorn, 2010). We implement these approaches with a linear, tree, or random forest model for the downstream predictive model. We treat the type of model as an hyper-parameter. We used the default parameter values for number of imputations and number of iterations in `mice`.

For linear predictors, the hyper-parameters are the Lasso penalty $\lambda$ and the amount of ridge regularization $\alpha$ (ElasticNet). For tree predictors, the hyper-parameter is the maximum depth. For the random forest predictors, the hyper-parameters are the maximum depth of each tree and the number of trees in the forest.

All hyper-parameters are cross-validated using a 5-fold cross-validation procedure on the training set.

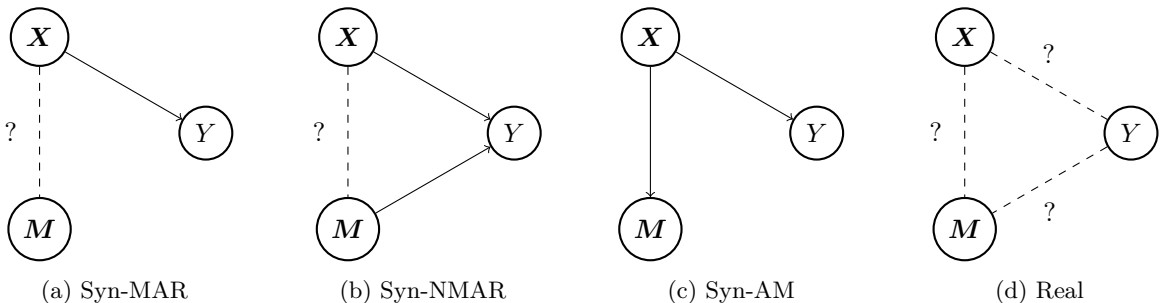

Figure C.1: Graphical representation of the 4 experimental designs implemented in our benchmark simulations with real-world design matrix $\boldsymbol{X}$. Solid (resp. dashed) lines correspond to correlations explicitly (resp. not explicitly) controlled in our experiments.

We report out-of-sample predictive power on the test set. For synthetic data, the test set consists of $5,000$ observations. For real data, we hold out 30% of the observations as a test set.

All experiments are replicated 10 times, with different (random) split into training/test sets between replications.

## D   Implementation of Impute-then-Regress Pipelines with MICE

To achieve the best achievable predictive power, Theorem 2.1 requires the imputation rule to be the same on the training set (on which the downstream predictive model is then trained) as on the test set. However, many imputation models (especially the best performing ones) do not impute missing values as a simple function of the observed features ($\mu(\boldsymbol{x}_{2:d})$ in the statement of Theorem 2.1) but rely on an iterative process where, at each iteration, current imputed values are used to train a new imputation model and then updated. Consequently, in practice, we cannot guarantee that the exact same imputation rule is used in training and testing.

We consider three implementations of impute-then-regress:

(V1) In the first variant, we simultaneously impute the train and test set, before training the model.

(V2) Secondly, we impute the training set alone, and then impute the test set with the *imputed* training data.

(V3) As a third option, we impute the training set alone and then the test set with the *original* training set.

Intuitively, (V1) should lead to the most consistent imputation across the training and test set but is not practical for predictive models that are meant to be used in production. Indeed, the behavior of the model on the test set should mimic its behavior on future observations, which, by definition, are unavailable (hence should not be used) at any stage of the calibration process. (V2) is the variant we compared mean-imputation against in Section 3.3. We intuit that (V3) will be less powerful than (V2) because the rules learned for imputing the test set might differ from the ones used for the training set.

We conduct a regression analysis (Table D.1), to assess the relative benefit of using (V1) and (V3) over (V2).

As expected, we observe that (V1) leads to higher accuracy than (V2) on the synthetic instances. On the semi-real and real data instances, however, we observe no strong statistically significant difference between the two variants. Regarding (V3) vs. (V2), when statistically significant (7 out of 11 cases), we find that the comparison is in favor of (V2). Henceforth, we recommend in practice to use (V2) since it mimics more closely the production pipeline and the theoretical requirements from Theorem 2.1.

Table D.1: Regression output for predicting the out-of-accuracy ($R^2$ or AUC) based on the implementation of the impute-then-regress method. We include dataset and $k_{misssing}$ fixed effects. We report regression coefficient values (and clustered standard errors).

| Design matrix $\boldsymbol{X}$ | Signal $Y$ | Missingness | V1 vs. V2 coeff. | V3 vs. V2 coeff. | Adjusted $R^2$ |
|---|---|---|---|---|---|
| Syn. | Syn. - Linear | MCAR | 0.0049 (0.0017)** | -0.0002 (0.0002) | 0.3483 |
| | | Censoring | 0.0047 (0.0014)*** | -0.0013 (0.0004)*** | 0.3484 |
| | Syn. - NN | MCAR | 0.0038 (0.0010)*** | 0.0 (0.0002) | 0.4289 |
| | | Censoring | 0.0052 (0.0023)*** | -0.0024 (0.0002)*** | 0.4920 |
| Real | Syn. - Linear | MAR | 0.0076 (0.0066) | -0.0067 (0.0024)** | 0.2325 |
| | | NMAR | 0.0020 (0.0030) | -0.0076 (0.0027)** | 0.3982 |
| | | AM | -0.0021 (0.0027) | -0.0017 (0.0025)*** | 0.3468 |
| | Syn. - NN | MAR | 0.0067 (0.0060) | -0.0028 (0.0012)* | 0.2805 |
| | | NMAR | 0.0030 (0.0036) | -0.0037 (0.0017)* | 0.3146 |
| | | AM | 0.0008 (0.0020) | -0.0005 (0.0015) | 0.3362 |
| Real | Real | Real | 0.0104 (0.0070) | -0.0016 (0.0020) | 0.8496 |

Controls: Dataset, $k_{missing}$ (if available); $p$-value: *:< 0.1, **:< 0.01, ***:< 0.001

# E  Additional Numerical Results

In this section, we report complementary numerical results to Sections 3-4.

In Table 1, we evaluate the benefit of mode imputation (vs. encoding missingness as a new category) for missing categorical variables on a collection of synthetic, semi-real, and real datasets. For the 34 real datasets, we consider 10 different training/test splits and compute the difference in accuracy between the two imputation strategies and conduct a paired t- and Wilcoxon test across all datasets to assess which method is more accurate (on average and more often). Aggregating differences in accuracy across all datasets might be fallacious because they cannot be considered as identically distributed. Actually, both tests were inconclusive on these datasets. Alternatively, we can do this analysis at a dataset level. For each dataset, we can compute the average difference in means across the 10 training/test splits and compute the associated $t$-statistics (or equivalently the associated z-score). Figure E.1 reports the distribution of the difference in mean accuracy (left panel) and of the z-scores across the 34 datasets. First, we observe that despite our best effort to aggregate accuracy metrics that are on the same scale ($2 \times AUC - 1$ for classification and $R^2$ for regression tasks), the difference in means are more concentrated for regression problems than for classification ones. Second, these distributions confirm the absence of a clear conclusion in these settings with almost as many datasets where mode imputation is clearly beneficial (difference in means strongly positive, z-score close to 0) as those where it is detrimental (difference in means strongly negative, z-score close to 1). On this regard, we should note that the mode imputation seems more detrimental for regression tasks than classification ones, although the limited number of datasets involve prevents us from drawing a strong conclusion (see Table E.1).

Table E.1: Difference in means and in pseudo-medians (with two-sided $p$-values) in out-of-sample accuracy from a $t$ and Wilcoxon test applied to assess the impact of mode imputation on downstream accuracy, on real datasets only. A negative value means that mode impute reduces accuracy compared with encoding missingn as a new category.

| Task | # comp. | $\Delta$ mean ($p$-value) | $\Delta$ pseudo-median ($p$-value) |
|---|---|---|---|
| Classification | 250 | 0.0118 (0.210) | 0.0000 (0.98) |
| Regression | 90 | -0.0049 (0.092) | -0.0002 (0.015) |

Figure E.2 reports the result of the same analysis for the comparison between `mice` and mean impute.

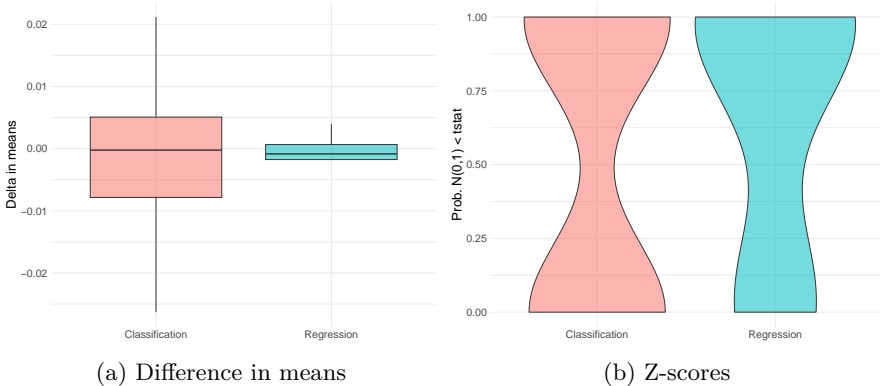

(a) Difference in means          (b) Z-scores

Figure E.1: Distribution of average difference in out-of-sample accuracy (left panel) and of the z-scores (right panel) across the 34 real datasets used to evaluate the effect of mode impute over encoding missingness as a new category. On the left panel, a negative value implies that mode imputation performs worse on average. On the right panel, a z-score close to 1 (resp. 0) implies that mode imputation has a detrimental (resp. beneficial) effect with strong confidence.

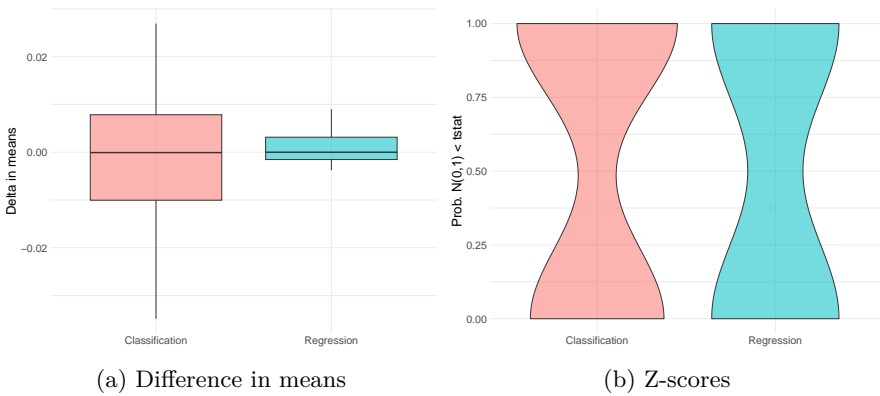

(a) Difference in means          (b) Z-scores

Figure E.2: Distribution of average difference in out-of-sample accuracy (left panel) and of the z-scores (right panel) across the 36 real datasets used to evaluate the effect of `mice` impute over mean impute. On the left panel, a negative value implies that `mice` performs worse on average. On the right panel, a z-score close to 1 (resp. 0) implies that `mice` has a detrimental (resp. beneficial) effect with strong confidence.

Figure E.3 complements Figure 1 by displaying the difference in accuracy between `mice`-then-regress and mean-impute-then-regress as both the number of observations and the proportion of missing entries vary. Unlike the proportion of missing entries, we do not observe a clear pattern in how sample size affects the relative performance of each approach.

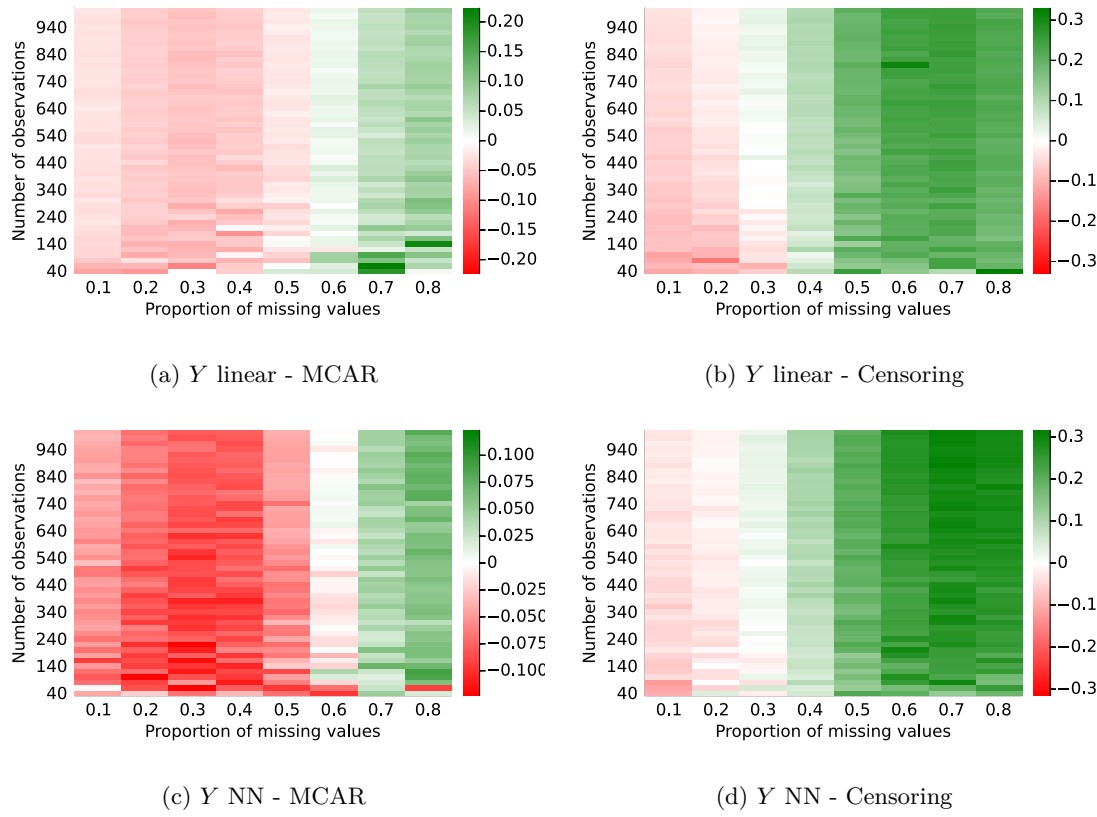

(a) $Y$ linear - MCAR

(b) $Y$ linear - Censoring

(c) $Y$ NN - MCAR

(d) $Y$ NN - Censoring

Figure E.3: Difference in out-of-sample $R^2$ between `mice`- and mean-impute-then-regress on fully synthetic data, as the proportion of missing entries and the sample size vary. A green/positive value indicates that mean impute is more accurate than `mice`.

Figure E.4 replicates Figure 3 for a random forest regressor instead. It reports the out-of-sample accuracy achieved by random forest as the number of observations $n$ increases, for three different treatment of missing data: no treatment, mean impute, and `mice`.

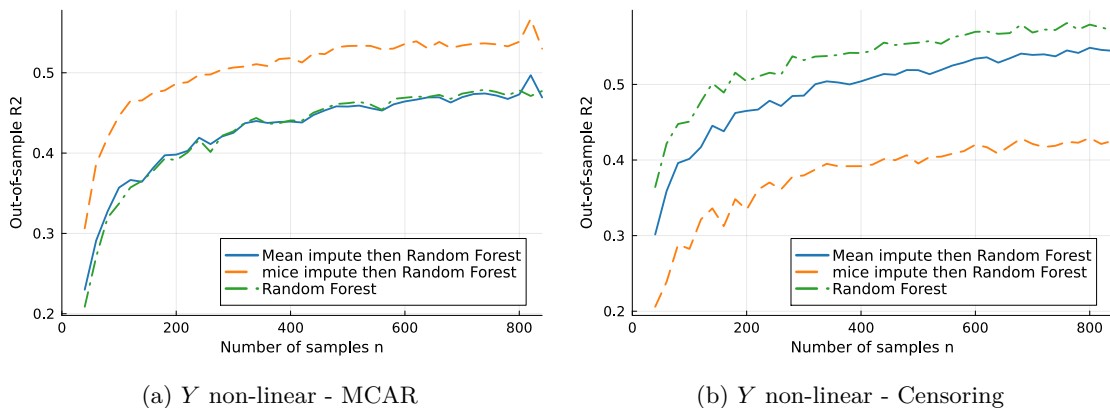

(a) $Y$ non-linear - MCAR

(b) $Y$ non-linear - Censoring

Figure E.4: Average out-of-sample $R^2$ of XGBoost with mean impute, `mice`, or no imputation method, on synthetic data with non-linear signal, NMAR missing data, and 40% of missing entries, as the number of samples $n$ increases. Results are averaged over 10 training/test splits.

