# OpenReview forum: "Simple Imputation Rules for Prediction with Missing Data: Theoretical Guarantees vs. Empirical Performance"
_TMLR — Accepted by TMLR_

### Review · Reviewer_Wsdy · 2024-04-10

**Summary Of Contributions:**

This paper discussed that mode-impute is asymptotically sub-optimal, while mean-impute is asymptotically optimal. The authors use theoretical proofs to support this statement and empirical study to demonstrate its correctness.

**Audience:**

Yes

**Claims And Evidence:**

Yes

**Requested Changes:**

Add Related Works. Add NNs as one type of predictive model. Add limitation discussion. Compare with the existing methods.

**Strengths And Weaknesses:**

Strengths:

1. The motivation is clear.

2. The paper is easy to follow.

3. The insights are significant.

Weaknesses:

1. It seems like the authors missed the related works. I suggest the authors add this section to provide an overview of the development of this research topic (i.e., Imputation).

2. For the experiment in Section 3.2, the authors designed the baselines (means and pseudo-medians). I am not very familiar with this domain. However, I suppose there are lots of established methods from existing papers on this topic. Why do the authors not compare with the existing baselines?

3. The predictive models are traditional machine learning models. No neural networks (NNs) are considered. The authors only use NNs to generate synthetic data. I would like to know the performance of advanced predictive models such as NNs.

4.  It is better to have a discussion about the limitations of this work.

---

> ### Author Response · Authors · 2024-04-23
> **Initial response to Reviewer Wsdy**
>
> Thank you for your careful reading of our paper and for assessing the significance and motivation of our analysis. Thank you also for suggesting some opportunities for improvement. We have simultaneously posted a preliminary revision on OpenReview to begin addressing your and other reviewers' comments, with edits indicated in blue.
>
> Regarding specific weaknesses/requested changes, which we number 1-4 in the order you mentioned them:
>
> 1. We are working to expand the literature review to more carefully describe the current state of imputation methods.
>
> 2. We would like to clarify what is meant by 'means' and 'pseudo-medians'. In Section 3.2, we consider two methods for imputing missing categorical variables. We obtain the performance of each method on each data set. We would like to support a statement of the form 'Method A achieves better accuracy than Method B'. Because the methods are applied exactly to the same instances, we can apply a paired test. If we apply a $t$-test, we are effectively testing whether 'Method A achieves better average accuracy than Method B' and the test statistics is the difference in means. If we apply a Wilcoxon test, we are testing whether `Method A achieves better accuracy than Method B most of the time' and the test statistics is called the pseudo-medians. The numbers reported in Table 1 under '$\Delta$ mean' and '$\Delta$ pseudo-median' correspond to the values of these test statistics, not baseline methods.
>
> 3. Currently, our experiments consider four different families of regressors (regularized linear regression, tree, random forest, XGBoost model). For the results presented in Section 3, the choice of the downstream regressor is done via 5-fold cross-validation (for each imputation strategy). We are currently assessing the code work and the time that would be required to run the experiments with neural networks as a fifth option. However, given that (i) we already consider highly non-linear regressors (i.e., RF and XGBoost), (ii) that the downstream regressor is chosen via cross-validation, and (iii) that were are dealing with fairly standard tabular datasets, we do not expect that adding neural network will significantly change the relative performance of each imputation strategy.
>
> 4. We have expanded the preamble of Section 4 to clearly delineate the theoretical and empirical limitations at a high level before diving into the details.

---

> ### Author Response · Authors · 2024-04-29
> **About Requested Change #3 (Adding Neural Networks)**
>
> Following your suggestion, we added a neural network regressor as one potential class of regressors (alongside regularized linear regression, decision tree, random forest, and XGBoost) and ran all the experiments with this additional regressor. All results (tables/figures) in the manuscript have been updated accordingly.

---

> ### Author Response · Authors · 2024-05-02
> **About Requested Changes #1 and #4**
>
> We now posted another revised version of the manuscript addressing concerns 1 and 4 more thoroughly.
> - Following your first suggestion, we have expanded the literature review on imputation methods.
> - Following your fourth suggestion, we have edited the discussion section to more clearly highlight the limitations.

---

### Review · Reviewer_PEFs · 2024-04-16

**Summary Of Contributions:**

This paper deals with the impute-then-regress framework, where missing data imputation is targeted to the downstream regression task. Unlike the pure missing data imputation task, the authors reveal that "crude" imputation works well in light of the downstream task. Specifically, Theorem 2.1 shows the form of the optimal predictor given by a universally consistent learning algorithm applied to imputed data, which is a convex combination of the imputed and non-imputed regressors. This optimal classifier becomes closer to the Bayes predictor in the form of Eq. (1) when it is easier to discern whether a given data is imputed. Eventually, the authors empirically tested this finding with the mode imputation for discrete data and mean imputation for continuous data, and the latter works well because the realization of imputed means would never precisely match the unobserved feature value (almost surely), i.e., detecting whether a given data is imputed or not is fairly easy.

**Audience:**

Yes

**Broader Impact Concerns:**

The current work mainly focuses on the theory of missing data analysis, and no ethical implications would be applicable.

**Claims And Evidence:**

Yes

**Requested Changes:**

The above comments in Weaknesses are expected to be addressed.

In the experiments, the following points can be clarified/improved.

+ In Table 1, it is not evident how 41 datasets used for "Real" are aggregated.
+ In the experiments in Section 3.2, "the out-of-sample accuracy [without] mode imputation" should be clarified. Is it simply regression from $\\boldsymbol{x}\_{2:d}$ to $y$?
+ In the experiments in Table 2, why not compare the mean imputation with a non-imputed regressor instead of the baseline mice? This comparison looks more natural because we want to see whether the mean imputation improves the downstream regressor over the vanilla regressor.

Below are minor suggestions.

+ In the third line on p.3, a space needs to be added at "For ease of exposition,[ ]we consider a simplified setting ..."
+ In Theorem 2.1, the hyphen in $f\_{\\mu-\\text{impute}}$ can be better typeset by "f\_{\mu\mathchar`-\text{impute}}".
+ Theorem/section headers in citations should be capitalized. For example, "Josse et al. (2019, [T]heorem 4)" in the first line of the last paragraph on p.4.
+ In 8th line in Section 3.1, "ampute" -> "impute"
+ In 11th line in Section 3.1, "UCI Machine Learning and RDatasets repositories" -> "UCI Machine Learning Repository"
+ In the 2nd line in Section 3.2, it should be better to have "for any $\\boldsymbol{x}$" for the statement "$p\_\\mu(\\boldsymbol{x}) > 0$ so $\\alpha(\\boldsymbol{x}) < 1$".

**Strengths And Weaknesses:**

## Strengths

+ **Free from the MAR assumption.** The main technical contribution, Theorem 2.1, is shown without the MAR assumption, a common assumption in the field of missing data imputation yet extremely challenging to verify in practice. The optimal classifier is derived without this assumption. The proof is based on a simple calculus and is not hard to follow.
+ *Large-scale experiments." The theoretical findings and discussions are tested with many real datasets. Though the performances are not as good as expected, the reason is carefully discussed.

## Weaknesses

+ **Unclear why $f\_{\\mu-\\text{impute}}$ is needed.** In Eq. (1), it is shown that all we need is to regress the observed covariates $\\boldsymbol{x}\_{2:d}$ to the outcome $y$ when $x\_1$ is missing ($m\_1=1$). This can be done without imputation by simply regressing $\\boldsymbol{x}\_{2:d} \\mapsto y$, and hence, the necessity of the intermediate imputation is unclear if we are interested solely in the downstream regression task. Additionally, I'm not sure whether the Bayes predictor in Eq. (1) can be derived straightforwardly from the general one shown in Eq. (3) in Le Morvan et al. (2021).
+ **Theory in Section 3.3.** The justification of the mean imputation hinges on the claim "$X\_1|\\boldsymbol{X}\_{2:d} = \\boldsymbol{x}\_{2:d}$, $M\_1 = 0$ is continuous [...] then, conditioned on $\\boldsymbol{X}\_{2:d} = \\boldsymbol{x}\_{2:d}$ and $M\_1 = 0$, the probability that $X\_1$ takes any specific value is $0$ so $p\_\\mu(\\boldsymbol{x}) = 0$ and $\\alpha(\\boldsymbol{x}) = 1$" (in the first paragraph in Section 3.3). While this is correct in the measure-theoretic viewpoint, does it mean any imputation methods can attain $\\alpha(\\boldsymbol{x}) = 1$? It seems that any method can avoid this measure-zero event $X\_1 = \\mu(\\boldsymbol{x}\_{2:d})$. This point can be discussed slightly more.

---

> ### Author Response · Authors · 2024-04-23
> **Initial response to Reviewer PEFs**
>
> Thank you for the thoughtful review, and particularly for highlighting that we do not rely on the MAR assumption for our analysis, and for noting our extensive numerical experiments using both real and synthetic data. Thank you also for suggesting potential improvements. We have simultaneously posted a preliminary revision on OpenReview to begin addressing your and other reviewers' comments, with edits indicated in blue.
>
> Regarding specific weaknesses, which we number 1-2 in the order you mentioned them:
>
> 1. You are right that in Eq. (1), we can simply achieve the Bayes optimal predictor by training two models, one for $m_1=1$ (using $\mathbf{x}_{2:d}$ only) and one for $m_1=0$ (using all the covariates, $\mathbf{x}$). However, in the general case where $1 \leq r \leq d$ features can be missing, this would lead to $2^r$ models to learn, which could be prohibitive. Hence, understanding cases where $\mu$-impute-then-regress can achieve the same performance is valuable.
>
> 2. We agree that in the case of continuous features, attaining $\alpha(\mathbf{x})=1$ is not difficult, even for a constant-imputation rule. This is an interesting consequence of our analysis --- conventional thinking would suggest that more accurate, or ``smarter'', imputation, should lead to better downstream predictions. But in fact, this is not the case asymptotically when the missing features are continuous.
>
> Regarding requested changes, which we number 1-3 in the order you mention them:
>
> 1. We are working on ways to clarify the results on real data.
>
> 2. The accuracy ``without mode imputation'' corresponds to the accuracy obtained when encoding missingness as its own category. We clarified this point in Section 3.2.
>
> 3. We do not fully understand this point. Could you please clarify what you mean by a ``vanilla non-imputed regressor''? Do you mean regressing on the features that are never missing (`complete feature' analysis)?
>
> We have also addressed all minor suggestions.

---

> ### Author Response · Authors · 2024-04-29
> **About Requested Change #1 (Table 1: How 41 datasets used for "Real" are aggregated?)**
>
> Thanks for offering us the opportunity to clarify this step.
> First, out of the 41 datasets, only 34 had a clearly identified target variable. For each of these 34 datasets, we considered 10 training/test split and compared mode imputation with encoding missingness as a new category for each of them, leading to 340 comparisons. We then conducted a paired t- and Wilcoxon test across all comparisons to assess which method is more accurate (on average and more often).
>
> We agree that aggregating differences in accuracy across all datasets might be fallacious because they cannot be considered as identically distributed. To alleviate this concern, we now conduct and report the same analysis at a dataset level (comparing only across the 10 training/test splits for each dataset) in Appendix E. We also report our results for classification and regression problems separately. These distributions confirm the absence of a clear conclusion in these settings with almost as many datasets where mode imputation is clearly beneficial as those where it is detrimental, but with relatively strong effect at a dataset level.
>
> We hope these additions clarify your concerns.

---

> > ### Comment · Reviewer_PEFs · 2024-05-05
> > **Response**
> >
> > Thank you for carefully addressing my comments.
> >
> > > We do not fully understand this point. Could you please clarify what you mean by a ``vanilla non-imputed regressor''? Do you mean regressing on the features that are never missing (`complete feature' analysis)?
> >
> > Initially, I looked at the experimental results in Table 1, which "compare the out-of-sample accuracy [...] with and without mode imputation" ("Results" in p.5 of the original version). This experiment looks reasonable to see the effect of the mode imputation, so I thought a similar comparison should be executed in Table 2 as well. However, I found that "without mode imputation" is unclear. In the updated manuscript, the corresponding sentence in p.5 has been updated as "that obtained when encoding missingness as its own category," which is not sufficiently clear. Can you clarify this first?

---

> > > ### Author Response · Authors · 2024-05-05
> > > **Clarification Table 1 "without mode imputation"**
> > >
> > > Thank you for following up.
> > > We agree that referring to Table 1 as evaluating "with or without mode imputation" was misleading because "without mode imputation" seemed to suggest that we do nothing regarding missing values in the latter approach, which is not the case. We always need to do something.
> > >
> > > In Section 3.2 (and Table 1), we evaluate the performance of mode imputation with the alternative of creating a new category corresponding to "missing". The latter approach is the one we are referring to as "encoding missingness as its own category". Is that clearer? If so, we will edit the manuscript accordingly.
> > >
> > > Similarly, in Section 3.3 (and Table 2), we evaluate the performance of mean impute with the alternative of a more sophisticated imputation model (i.e., mice in our experiments) to illustrate that, as theory suggests, the simplicity of mean impute does not translate into systematic under-performance.

---

> > > > ### Comment · Reviewer_PEFs · 2024-05-20
> > > > **Response**
> > > >
> > > > > In Section 3.2 (and Table 1), we evaluate the performance of mode imputation with the alternative of creating a new category corresponding to "missing". The latter approach is the one we are referring to as "encoding missingness as its own category". Is that clearer? If so, we will edit the manuscript accordingly.
> > > >
> > > > It's fine now. Thanks for your elaboration.

---

### Review · Reviewer_o3iN · 2024-04-20

**Summary Of Contributions:**

The authors study imputation of missing data. They first report a theoretical result which relates the outputs of an imputed-trained predictor to various expectations. An implication of this result is that the imputed-trained predictor will be consistent (expected output equal to the true expected value for a given input, IIUC) under certain conditions, which mostly relate to how well the predictor can detect missingness and imputation..

Experiments seem to confirm the implications of the theory. For discrete data, mode-imputation is shown to reduce accuracy in comparison to indicating missingness with a separate category. For continuous data, mean-inputation (which effectively indicates missingness under mild assumption) is competitive with a much more complex method, at least when much data is missing.

**Audience:**

Yes

**Broader Impact Concerns:**

I do not see any broader impacts.

**Claims And Evidence:**

Yes

**Requested Changes:**

- What, exactly is the proposed training method? In Theorem 2.1 we read "systematically imputing" x1. Does this apply to all x1, or only to the missing one? That is, do we replace the non-missing x1 with their imputed values in the training set or not?

- Also in this theorem, the first equation seems to suggest that the method will produce the correct result when mu != x1, i.e. precisely when the imputation is incorrect?? Is that actually the case?

- When the imputation is incorrect, the output will be a weighted mean of the value predicted while ignoring x1, and trhe value predicted while inputing x1? But isn't the latter exactly what we are doing?

- Basically please just explain carefully what is going on there and what the terms actually mean.

- In 3.3, to be clear, the reasoning would apply to any imputation method resulting in values outside the distribution of the true non-missing X1's? Even simply mu = +inf for any input would produce "consistent" results?

**Strengths And Weaknesses:**

Strengths: The problem is important, the results seem interesting, and the experiments seem to confirm the suggested interpretation.

Weaknesses: The paper is extremely confusing. The exact training method is ambiguous, and the explanations of the theoretical results are incomprehensible.

---

> ### Author Response · Authors · 2024-04-23
> **Initial response to Reviewer o3iN**
>
> We thank you for your careful reading of our paper and for assessing our work as important, interesting, and empirically validated. We are also grateful for the suggested improvement directions, in particular around the clarity of the exposition around Theorem 2.1. We have simultaneously posted a preliminary revision on OpenReview to begin addressing your and other reviewers' comments, with edits indicated in blue.
>
> Regarding your specific requested changes, which we number 1-5 in the order you asked them:
>
> 1. We are only imputing $x_1$ when it is missing, we do not alter non-missing values. We clarified this point in the revision.
>
> 2. Actually, $\mu \neq x_1$ does not mean that the imputation is incorrect. The predictor $f_{\mu-impute}$ applies to a vector $\mathbf{x}$ after imputation. $\mu \neq x_1$ means that the value of the feature $x_1$ for that observation $\mathbf{x}$ is different from the value that would be typically imputed for such a vector (given $\mathbf{x}_{2:d}$). In other words, it is safe to assume that for this vector $x_1$ was not missing ($m_1=0$) and has not been imputed (otherwise it would precisely be equal to $\mu$). We have clarified this in the discussion immediately before and after Theorem 2.1.
>
> 3. Your interpretation of Theorem 2.1 when $\mu=x_1$ is almost correct. More precisely, the second term in the second case (weighted by $1-\alpha(\mathbf{x})$) is the conditional expectation over all points for which $x_1$ is not missing and happens to precisely equal the value $\mu$, i.e., the value we \emph{would} have used to impute it if it \emph{had} been missing.
>
> 4. As detailed in points 1-3, we have significantly updated the discussion around Theorem 2.1. Please let us know if there are ways we can enhance clarity even further.
>
> 5. Yes, your understanding is correct. We have clarified this point in our revision.

---

### Decision · Action_Editor_hn2g · 2024-05-28

**Recommendation:** Accept as is

**Comment:**

The paper analyzed the impute-then-regress framework, where the missing data imputation is performed for the downstream regression task.  Validity of the theory is empirically assessed on artificial and real datasets.

The reviewers found significant contributions with criticisms on the clarity, which have been well addressed by the authors.

**Audience:**

yes

**Claims And Evidence:**

yes